# TMEM25 is a Par3-binding protein that attenuates claudin assembly during tight junction development

Sachiko Kamakura, Junya Hayase, Akira Kohda, Yuko Iwakiri, Kanako Chishiki, Tomoko Izaki & Hideki Sumimoto [ID] ✉

## Abstract

The tight junction (TJ) in epithelial cells is formed by integral membrane proteins and cytoplasmic scaffolding proteins. The former contains the claudin family proteins with four transmembrane segments, while the latter includes Par3, a PDZ domain-containing adaptor that organizes TJ formation. Here we show the single membrane-spanning protein TMEM25 localizes to TJs in epithelial cells and binds to Par3 via a PDZ-mediated interaction with its C-terminal cytoplasmic tail. TJ development during epithelial cell polarization is accelerated by depletion of TMEM25, and delayed by overexpression of TMEM25 but not by that of a C-terminally deleted protein, indicating a regulatory role of TMEM25. TMEM25 associates via its N-terminal extracellular domain with claudin-1 and claudin-2 to suppress their *cis*- and *trans*-oligomerizations, both of which participate in TJ strand formation. Furthermore, Par3 attenuates TMEM25–claudin association via binding to TMEM25, implying its ability to affect claudin oligomerization. Thus, the TJ protein TMEM25 appears to negatively regulate claudin assembly in TJ formation, which regulation is modulated by its interaction with Par3.

**Keywords** Claudin; Epithelial Cell Polarity; Par3; Tight Junction
**Subject Category** Cell Adhesion, Polarity & Cytoskeleton

## Introduction

In multicellular organisms, epithelial cell sheets play an essential role in the separation of the internal tissue compartments from their external environments. The integrity of the sheets in vertebrates is maintained by cell–cell adhesion junctions, comprising tight junctions (TJs), adherens junctions, and desmosomes. TJs localize to the most apical part of lateral membranes in epithelial cells and function as a barrier and a channel that regulates the paracellular permeation of ions and solutes (Tsukita et al, 2001; Anderson and Van Itallie, 2009; Otani and Furuse, 2020; Piontek et al, 2020). As TJs also act as a fence to restrict the diffusion of lipids and proteins between the apical and basolateral domains of the plasma membrane, TJ assembly is thought to be closely linked to the establishment and maintenance of apico-basal cell polarity (Zihni et al, 2016; Riga et al, 2020).

TJs consist of a meshwork of linear intramembrane stands (TJ strands) constituted by a variety of integral membrane proteins: the tetra-span transmembrane proteins such as claudins and occludin and the single-membrane-spanning protein JAM-A of the immunoglobulin (Ig) superfamily, being in contact with soluble adaptor proteins including ZO-1, ZO-2, and Par3 (Tsukita et al, 2001; Zihni et al, 2016). The backbone of the TJ strands is formed by claudins of a multigene family with 27 members in mammals (Tsukita et al, 2019): indeed, in canine kidney epithelial MDCK cells, knockout of the five major claudins expressed in these cells (claudin-1, -2, -3, -4, and -7) results in a loss of TJ strand formation (Otani et al, 2019). In TJ strands claudins assemble in a side-by-side (*cis*) and an intercellular head-to-head (*trans*) manner. Although both *cis*- and *trans*-interactions of claudins are critical for TJ strand formation (Suzuki et al, 2015; Tsukita et al, 2019; Piontek et al, 2020), the regulatory mechanisms for claudin assembly have remained largely unclear.

The control of TJ formation involves the evolutionarily conserved scaffold protein Par3, which is crucial for polarity establishment of *C. elegans* zygotes and *Drosophila* oocytes (Suzuki and Ohno, 2006; St Johnston, 2018). The cell polarity protein Par3 contains the N-terminal conserved region 1 (CR1) for self-oligomerization, three copies of the PDZ domain for protein–protein interaction, and the CR3 containing the aPKC-binding region (St Johnston and Ahringer, 2010; Rouaud et al, 2020; Fig. 1A). At the onset of mammalian epithelial cell polarization, Par3 forms a complex of the Par6–aPKC heterodimer and is recruited at least in part via the 1st PDZ domain (PDZ1)-mediated interaction with JAM-A to the primordial cell–cell junction, which also contains the TJ protein ZO-1 and the adherens junction proteins E-cadherin and nectins (Zihni et al, 2016; Steinbacher et al, 2018). The Par3–Par6–aPKC complex recruited is considered to act as a platform for TJ assembly and apical domain development (Suzuki et al, 2001; 2002; Chen and Macara, 2005; Horikoshi et al, 2009). However, it has not been fully understood about the molecular mechanism whereby Par3 organizes TJ formation, especially the impact of Par3 in claudin assembly.

In the present study, we have identified the single-span Ig family protein TMEM25 (Katoh and Katoh, 2004) as a novel Par3-binding protein. TMEM25, localizing to the TJ in epithelial cells, directly

Department of Biochemistry, Kyushu University Graduate School of Medical Sciences, Fukuoka, Japan. ✉E-mail: sumimoto.hideki.851@m.kyushu-u.ac.jp

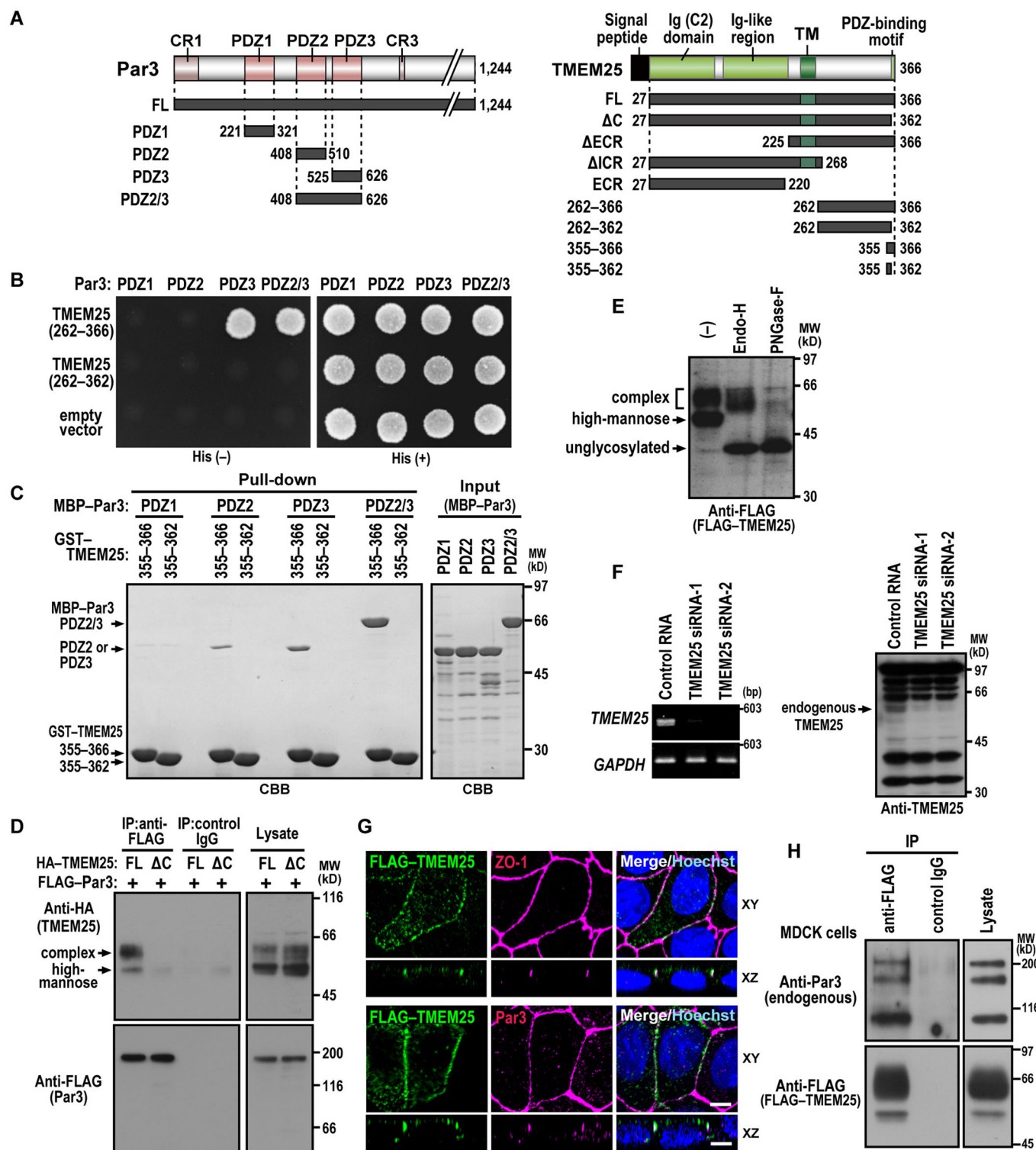

interacts via its C-terminal PDZ-binding motif with Par3-PDZ3 and also with Par3-PDZ2, but to a lesser extent. The role of TMEM25 as a regulator of TJ formation is indicated by the following findings: TJ development in kidney epithelial MDCK cells is accelerated by depletion of TMEM25, whereas it is prevented by overexpression of TMEM25 but not by that of a mutant protein lacking the C-terminal tail. TMEM25 specifically binds to claudin-1 and claudin-2 and inhibits claudin assembly in both *cis*- and *trans*-manners. Furthermore, TMEM25 binding to claudin is attenuated by Par3 in a manner dependent on the C-terminal PDZ-binding motif, suggesting that Par3 regulates claudin assembly via interacting with TMEM25.

◀ **Figure 1. Interaction of TMEM25 with Par3.**

(A) Schematic representation of constructs for full (FL) or various lengths of Par3 and TMEM25. CR, conserved region; PDZ, PSD-95/DLG/ZO-1; Ig, Immunoglobulin; TM, transmembrane; ECR, extracellular region; and ICR, intracellular region. (B) The yeast two-hybrid interaction between the intracellular region of TMEM25 and the PDZ domains of Par3. HF7c cells transformed with a pair of pGADGH encoding TMEM25 (amino acid residues 262–366 or 262–362) and pGBK-Par3 (PDZ1, PDZ2, PDZ3, or PDZ2/3) were grown in the presence (+) or absence (−) of histidine (His). (C) Direct interaction between the C-terminus of TMEM25 and the PDZ domains of Par3. GST–TMEM25 (355–366 or 355–362) was incubated with MBP–Par3 (PDZ1, PDZ2, PDZ3, or PDZ2/3), and pulled down with glutathione-Sepharose-4B beads, followed by SDS-PAGE analysis with *Coomassie Brilliant Blue* (CBB) staining. Positions for marker proteins are indicated in kilodaltons (kD). (D) TMEM25–Par3 interaction in mammalian cells. FLAG-Par3-FL was co-expressed with HA–TMEM25-FL or HA–TMEM25-ΔC in HEK293 cells, and proteins in the cell lysate were immunoprecipitated (IP) with the anti-FLAG M2 monoclonal antibody or control IgG, followed by immunoblot analysis with the indicated antibodies. (E) Glycosylation of TMEM25. Proteins in the lysate from MDCK cells expressing FLAG-TMEM25-FL were extracted with Triton X-100, and digested with Endo-H or PNGase-F. Proteins were analyzed by immunoblot with the anti-FLAG antibody. (F) Expression of TMEM25 in MDCK cells. MDCK cells transfected with negative control RNA or TMEM25-specific siRNA (TMEM25 siRNA-1 or siRNA-2) were analyzed by RT-PCR (left) or immunoblot with the anti-TMEM25 antibody (right). Positions for marker fragments of DNA and marker proteins are indicated in base pair (bp) and kilodaltons (kD), respectively. (G) Localization of TMEM25 to the TJ. Shown are subcellular localization of FLAG-tagged TMEM25 induced by the Tet-on system in MDCK cells. Cells transfected with the pTetOne-FLAG–TMEM25-FL expression plasmid were treated with 1 µg/ml doxycycline for 48 h, and then fixed and stained as indicated. Cross-sectional z-stack analysis (xz) is also shown. Scale bars, 5 µm. (H) The endogenous Par3 interacts with TMEM25 expressed in MDCK cells. Proteins in the lysate of MDCK cells expressing FLAG–TMEM25-FL were immunoprecipitated (IP) with the anti-FLAG M2 monoclonal antibody or control IgG, followed by immunoblot analysis with the indicated antibodies. Source data are available online for this figure.

# Results

## TMEM25 directly interacts with Par3

To identify a Par3-interacting protein that regulates TJ formation, we screened a human fetal cDNA library in the yeast two-hybrid system using a region containing the second and third PDZ domains (PDZ2/3) of human Par3 as a bait (Fig. 1A). A positive clone obtained encodes an intracellular region of amino acid residues 276–366 in the type I transmembrane protein TMEM25 (Katoh and Katoh, 2004). Human TMEM25 of 366 amino acids comprises an N-terminal signal peptide sequence, a typical C2-type immunoglobulin (Ig) domain, a region of an Ig-like fold, a single transmembrane segment, and an intracellular region (ICR) with the extreme C-terminal tetrapeptide Glu-Ile-Trp-Leu (amino acids 363–366) (Figs. 1A and EV1), which corresponds to a class II PDZ-binding motif (Hung and Sheng, 2002).

The C-terminal ICR of TMEM25 exhibited a two-hybrid interaction with PDZ3 as well as PDZ2/3 but not with PDZ1 or PDZ2 among the three PDZ domains of Par3 (Fig. 1A,B). Using glutathione *S*-transferase (GST)- and maltose-binding protein (MBP)-fused proteins that were bacterially expressed and affinity-purified, we found that GST–TMEM25-(355–366) was effectively pulled down with MBP–Par3-PDZ2/3, indicative of a direct interaction between TMEM25 and Par3 (Fig. 1C). The ICR of TMEM25 interacted with Par3-PDZ3 and also with Par3-PDZ2, but to a lesser extent, whereas it failed to bind to Par3-PDZ1 (Fig. 1C). Truncation of the C-terminal tetrapeptide of TMEM25 resulted in a loss of the interaction with Par3 in both the yeast two-hybrid system and the GST pull-down assay (Fig. 1B,C), highlighting a crucial role of the C-terminal PDZ-binding motif. When expressed as full-length (FL) protein in HEK293 cells, FLAG–Par3-FL co-precipitated with HA–TMEM25-FL but not with a mutant protein without the C-terminal tetrapeptide (HA–TMEM25-ΔC) (Fig. 1D). These findings indicate that TMEM25 directly binds to Par3 via PDZ-mediated interaction.

## TMEM25 is a glycoprotein that localizes to TJs

TMEM25 has six putative *N*-glycosylation sites of Asn-x-Ser/Thr motif (x is any amino acid but not proline) (Breitling and Aebi, 2013) in the extracellular region (ECR) (Fig EV1). Indeed TMEM25, expressed as a

FLAG-tagged protein in MDCK cells, was sensitive to treatment with peptide:*N*-glycosidase F (PNGase-F) and endoglycosidase H (Endo-H) (Fig. 1E), confirming that TMEM25 undergoes *N*-glycosylation: the upper and lower bands on SDS-PAGE contained (Endo-H-resistant) complex-type and (Endo-H-sensitive) high-mannose-type *N*-glycans, respectively. When expressed in HEK293 cells, Par3 preferentially interacted with TMEM25 bearing the complex-type *N*-glycans (Fig. 1D), a mature sugar chain of proteins that exist at the plasma membrane (Stanley, 2011), suggesting that the interaction mainly occurs at the plasma membrane. Indeed, Par3 and TMEM25 co-localized to the plasma membrane in HEK293 cells (Fig. EV2A).

The RNA-Seq data from the Ensembl database (www.ensemble.org) reveals that the *TMEM25* transcript is abundant in the brain and epithelial organs such as the kidney and intestine. Consistent with this, *TMEM25* mRNA was detected in the canine kidney epithelial line MDCK cells by RT-PCR analysis (Fig. 1F), and the TMEM25 protein was present in MDCK cells (Fig. 1F) and human intestinal Caco-2 cells (Fig. EV3A) as shown by immunoblot analysis. To investigate the subcellular localization of TMEM25, we expressed FLAG–TMEM25 in MDCK and Caco-2 cells and stained it with the anti-FLAG antibody, because no available antibodies were suitable for staining endogenous TMEM25. FLAG–TMEM25 predominantly accumulated at TJs as defined by the TJ-associated adaptor protein ZO-1 in MDCK (Fig. 1G) and Caco-2 (Fig. EV3B) cells, when the TMEM25 protein was expressed at a low level using the Tet-on inducible expression system that allows for control of gene expression in a manner dependent on the dose of the tetracycline analog doxycycline (Dox) (Gossen and Bujard, 1992). In this experiment, after the formation of cell–cell contacts, we carefully induced a low expression of FLAG–TMEM25 by the addition of 0.2–1 µg/ml of Dox, because its high expression led to an impaired TJ formation as described later. FLAG–TMEM25 effectively interacted with endogenous Par3 in MDCK cells (Fig. 1H) and co-localized with Par3 at TJs in MDCK cells (Fig. 1G), suggesting that TMEM25–Par3 interaction occurs at the junctional region. Thus, TMEM25 appears to be a membrane-integrated glycoprotein that accumulates at TJs in mammalian epithelial cells.

## TMEM25 self-assembles in *cis* but not in *trans*

JAM-A is also a single membrane-spanning TJ glycoprotein of the Ig superfamily; it contains a membrane-distal V-type and a

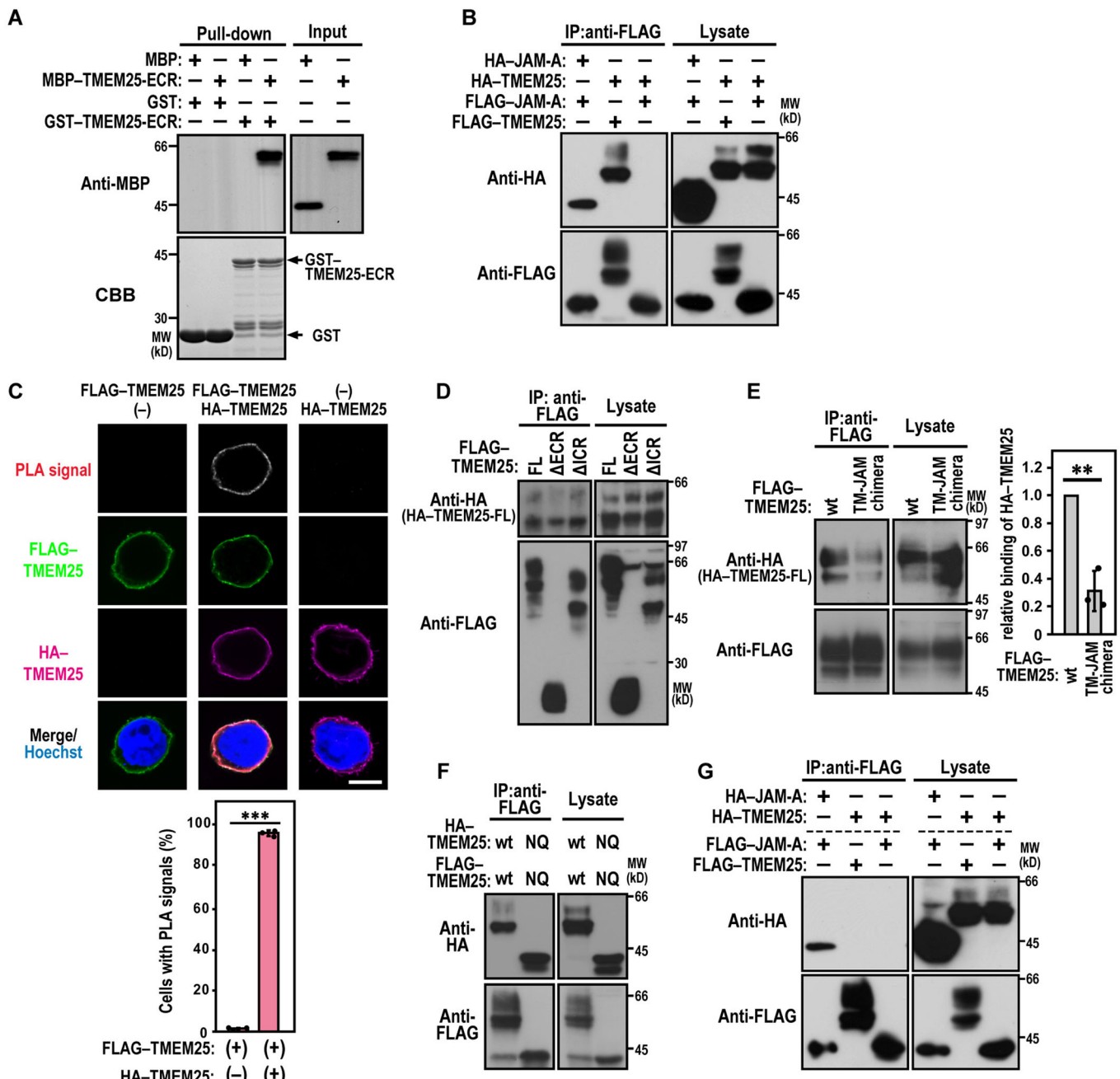

membrane-proximal C2-type Ig domain in the ECR and is known to form a homo-dimer (Ebnet, 2017). To test whether TMEM25 also self-associates, we purified GST- and MBP-tagged TMEM25-ECR and used them in a pull-down assay with glutathione-Sepharose beads. As shown in Fig. 2A, GST–TMEM25-ECR efficiently bound to MBP–TMEM25-ECR but not to MBP alone, suggesting a direct homophilic interaction of TMEM25-ECR. Self-association of full-length TMEM25 was also observed by co-immunoprecipitation analysis using HEK293 cells with simultaneous expression of FLAG- and HA-tagged TMEM25: both proteins were co-precipitated with an anti-FLAG antibody (Fig. 2B). To clarify whether TMEM25 self-associates in a side-by-side (*cis*) manner, we analyzed the homophilic interaction in a single cell via

the proximity ligation assay (PLA). As shown in Fig. 2C, the PLA signal was detected on the cell surface solely in a HEK293 cell expressing both HA–TMEM25 and FLAG–TMEM25 but not in a cell with either protein, under the conditions where the anti-HA and anti-FLAG antibodies were simultaneously used as primary antibodies (Fig. 2C). It is thus likely that TMEM25 self-associates in a *cis* manner at the plasma membrane.

The self-association of TMEM25 appears to involve the ECR and the transmembrane segment, because truncation of the ECR but not that of the ICR resulted in a slightly impaired interaction with TMEM25-FL (Fig. 2D) and the interaction was also attenuated by replacement of the transmembrane segment with the corresponding region of JAM-A (Fig. 2E), a protein incapable of binding to TMEM25

**Figure 2.  TMEM25 homophilic interaction in *cis* and in *trans*.**

(A) Direct homophilic interaction of TMEM25-ECR. GST–TMEM25-ECR (amino acid residues 27–220) or GST alone was incubated with MBP–TMEM25-ECR or MBP alone. Proteins pulled down with glutathione-Sepharose-4B beads were subjected to SDS-PAGE, and stained with *Coomassie Brilliant Blue* (CBB) or analyzed by immunoblot with the anti-MBP antibody. Positions for marker proteins are indicated in kilodaltons (kD). (B) Homophilic *cis*-interaction of TMEM25. HEK293 cells co-expressing FLAG–TMEM25 or FLAG–JAM-A with HA–TMEM25 or HA–JAM-A, respectively, were cultured at low density, and proteins in the cell lysate were immunoprecipitated (IP) with the anti-FLAG M2 antibody, followed by immunoblot analysis with the indicated antibodies. (C) The PLA for detection of *cis*-interaction of TMEM25. A HEK293 cell expressing FLAG–TMEM25 and/or HA–TMEM25 was fixed and subjected to PLA using the anti-FLAG rabbit polyclonal and anti-HA mouse monoclonal antibodies as primary antibodies. Bar graphs indicate quantification of cells with PLA signals at the plasma membrane. Values are means ± S.D. from three independent experiments ($n \geq 110$ cells/ experiment). ***$P < 0.001$ (Student's *t*-test). Scale bar, 10 μm. (D, E) Regions involved in homophilic *cis*-interaction of TMEM25. HA–TMEM25-FL was co-expressed in HEK293 cells with various lengths of FLAG–TMEM25 (D) or with FLAG-tagged TM-JAM chimera, a mutant TMEM25 protein with the replacement of its transmembrane segment with the corresponding region of JAM-A (E). The cells were cultured at low density, and proteins in the cell lysate were immunoprecipitated (IP) with the anti-FLAG M2 antibody, followed by immunoblot analysis with the indicated antibodies. Bar graphs in **E** indicate the relative ratios of HA–TMEM25 to FLAG–TMEM25 in the precipitates; the ratio of HA–TMEM25 to FLAG–TMEM25 (wt) is set as 1.0. Values are means ± S.D. from three independent experiments. **$P < 0.01$ (Student's *t*-test). FL, full-length; ECR, extracellular region; and ICR, intracellular region. (F) Role of *N*-glycosylation in homophilic interaction of TMEM25. FLAG–TMEM25 (wt or NQ) were co-expressed with HA–TMEM25 (wt or NQ) in HEK293 cells. Proteins in the cell lysate were immunoprecipitated (IP) and analyzed as in (C). wt, wild-type; and NQ, a mutation leading to the N106Q/N119Q/N162Q/N175Q/N192Q/N205Q substitution. (G) Homophilic *trans*-interaction of TMEM25. HEK293 cells expressing a FLAG-tagged TMEM25 (or JAM-A) were co-cultured with the cells expressing an HA-tagged TMEM25 (or JAM-A) to confluency. Co-cultured cells were lysed, and proteins in the lysate were immunoprecipitated (IP) with the anti-FLAG antibody, followed by immunoblot analysis with the indicated antibody. Source data are available online for this figure.

(Fig. 2B). We next examined the role of *N*-glycosylation in TMEM25–TMEM25 interaction by introducing glutamine substitution for all the six asparagine residues to be glycosylated: $Asn^{106}$, $Asn^{119}$, $Asn^{162}$, $Asn^{175}$, $Asn^{192}$, and $Asn^{205}$. As shown in Fig. 2F, the substitution did not affect the *cis*-interaction, suggesting that TMEM25 self-assembled in a manner independent of *N*-glycosylation, which agrees with the ability of bacterially expressed (i.e., non-glycosylated) TMEM25-ECR to self-associate (Fig. 2A).

It is well known that JAM-A forms a homo-dimer not only in a *cis* manner but also in an intercellular head-to-head (*trans*) manner to directly contribute to cell–cell contact (Ebnet et al, 2000; Kostrewa et al, 2001; Prota et al, 2003). In contrast, TMEM25 failed to self-associate in a *trans*-configuration: homophilic interaction was not observed when FLAG–TMEM25-expressing HEK293 cells and HA–TMEM25-expressing HEK293 cells were co-cultured to confluence and proteins in the cell lysate were precipitated by the anti-FLAG antibody (Fig. 2G). Under the same experimental conditions, as expected, JAM-A self-assembled in *trans* (Fig. 2G) as well as in *cis* (Fig. 2B). The failure of TMEM25 to self-associate in *trans* suggests that this protein does not contribute directly to cell–cell contact. Of note, TMEM25 was correctly presented on the surface of HEK293 cells within confluent monolayers as well as those in subconfluent cultures (Fig. EV2B), whereas TMEM25 localized to the TJ in polarized MDCK cells (Fig. EV2B).

## TMEM25 negatively regulates TJ development

To test the possibility that the Par3-binding protein TMEM25 plays a role in TJ formation, we performed a $Ca^{2+}$ switch assay using TMEM25-depleted MDCK cells: cells were depolarized by culture in low $Ca^{2+}$ media, and then TJs were re-assembled by switch to $Ca^{2+}$-rich media. TJ development can be monitored by measuring the transepithelial electrical resistance (TER), a functional measure of TJ integrity (Matter and Balda, 2003). As shown in Fig. 3A, $Ca^{2+}$-induced increase in TER of TMEM25-depleted MDCK cells was facilitated compared with that of control cells, indicative of an accelerated TJ development. In contrast, under the present conditions, TER increase was delayed in Par3-depleted cells (Fig. 3A), which agrees with previous observations (Chen and Macara, 2005; Horikoshi et al, 2009). Depletion of TMEM25 in MDCK cells also facilitated $Ca^{2+}$-induced

decline in paracellular leakage of 4-kDa fluorescein-isothiocyanate (FITC)-conjugated dextran (Fig. 3B), further supporting the idea that TJ development is accelerated in TMEM25-depleted cells.

To confirm the role of TMEM25 in TJ formation, we performed immunostaining for TJ proteins in TMEM25-depleted cells. At 2 h after $Ca^{2+}$ switch, the TJ adaptor protein ZO-1 (Fig. 3C) as well as Par3 (Fig. EV4A) localized to cell–cell contact sites in a discontinuous manner in control cells. On the other hand, TMEM25 depletion facilitated the formation of continuous junctions with the correct lateral localization of the TJ protein ZO-1 and the AJ protein E-cadherin (Figs. 3C and EV4A), indicating an accelerated TJ development in cells that maintain apico-basal polarity. Accumulation of TJ membrane-integrated proteins, such as occludin, claudin-1, and claudin-2, was also accelerated in TMEM25-depleted cells (Fig. 3D,E). Similarly, TMEM25 knockdown in Caco-2 cells led to acceleration of TJ assembly (Fig. EV3C). These findings indicate that TMEM25 serves as a negative regulator of TJ development.

When FLAG-tagged, full-length TMEM25 (FLAG–TMEM25-FL) was highly expressed using the Tet-on system by treatment of MDCK cells with 5 μg/ml of Dox, assembly of the TJ proteins ZO-1 (Fig. 4A) and Par3 (Fig. EV4B) after $Ca^{2+}$ switch was severely delayed as estimated by immunostaining, which agrees with the negative role of TMEM25 in TJ development. On the other hand, over-expression of TMEM25-ΔC, which lacks the C-terminal four amino acid residues for interaction with PDZ domains, did not affect TJ protein assembly in MDCK cells (Fig. 4A). Furthermore, to examine TJ development at the population level by measuring the TER, we expressed FLAG–TMEM25 proteins in MDCK cells via the lentiviral transfection system. As shown in Fig. 4B, $Ca^{2+}$-induced increase in TER was delayed in TMEM25-FL-expressing cells but not in TMEM25-ΔC-expressing cells. Thus, TMEM25 appears to negatively regulate TJ development in a manner dependent on a PDZ domain-mediated interaction.

## TMEM25 is required for correct cyst formation in 3D-cultured MDCK cells

The present finding that TMEM25 depletion in MDCK cells facilitates TJ development in the $Ca^{2+}$ switch assay (Fig. 3) suggests

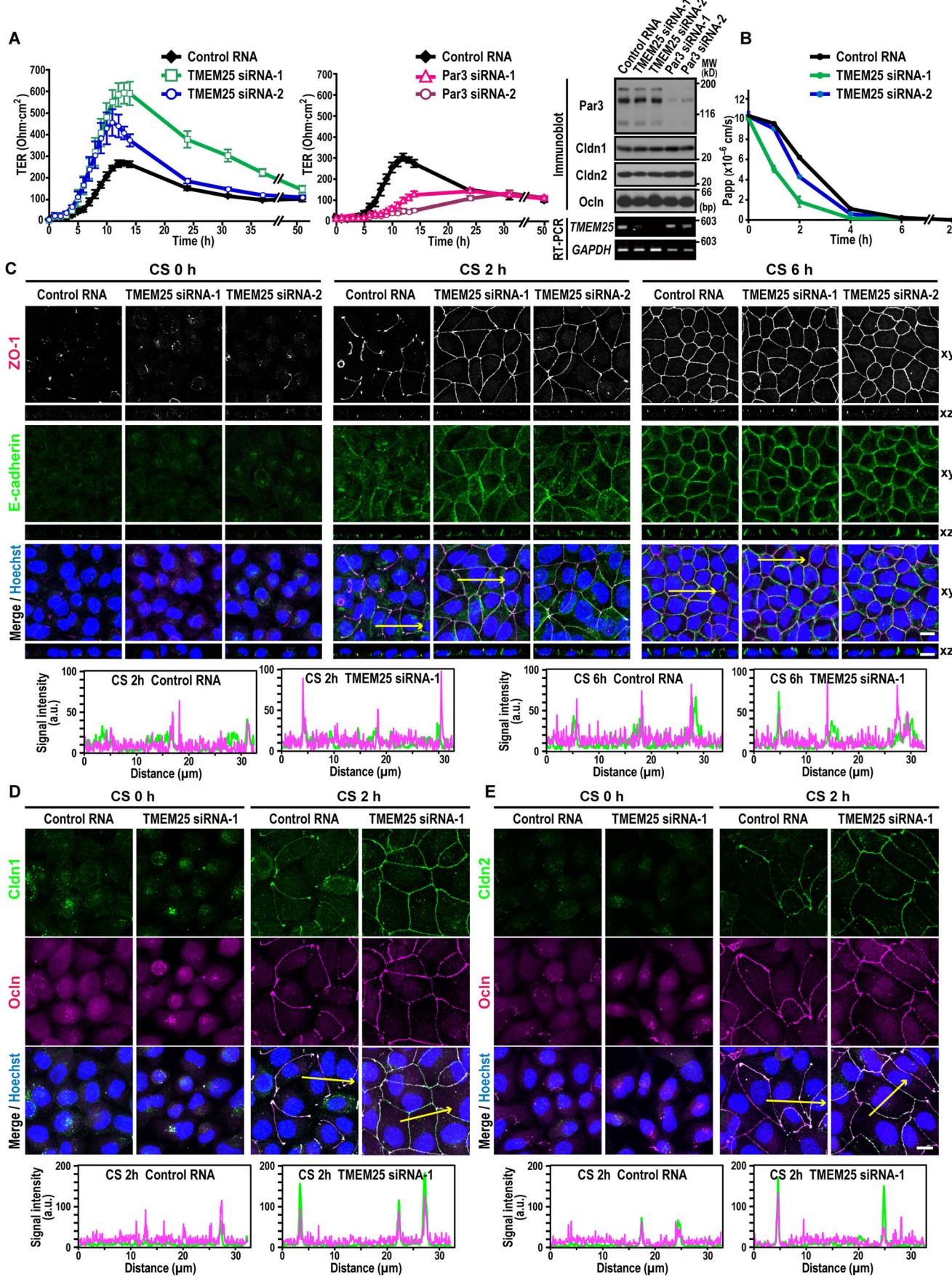

**Figure 3. Effect of TMEM25 depletion on TJ development.**

(A) $Ca^{2+}$-induced TER changes of monolayers of TMEM25-depleted MDCK cells. TER of monolayers of TMEM25- or Par3-depleted MDCK cells was monitored after $Ca^{2+}$ switch (left and middle panels): values are means ± S.D. from three independent experiments. The effect of transfection with TMEM25 siRNA or Par3 siRNA was analyzed by immunoblot with the indicated antibodies and by RT-PCR for expression of mRNA for *TMEM25* and *GAPDH* (right panel): positions for marker proteins or marker DNA fragments are indicated in kilodaltons (kD) or base pair (bp), respectively. (B) The paracellular flux of FITC-dextran in TMEM25-depleted MDCK cells after $Ca^{2+}$ switch. At the indicated time point after $Ca^{2+}$ switch, 4-kDa FITC-dextran was added to the apical compartment of the Transwell chamber and incubated for 30 min. From the basal compartment of the chamber, the culture medium was collected and the fluorescence intensity was measured. Values are means ± S.D. from three independent experiments. (C–E) Localization of TJ proteins in TMEM25-depleted MDCK cells after $Ca^{2+}$ switch: ZO-1 (C); claudin-1 (Cldn1) and occludin (Ocln) (D); and claudin-2 (Cldn2) and occludin (Ocln) (E). Shown are representative images of TMEM25-depleted MDCK cells after $Ca^{2+}$ switch (upper). Cells were fixed 0 h, 2 h, or 6 h after $Ca^{2+}$ switch (CS) and stained with the indicated antibodies and Hoechst. Confocal images of the cells were stacked along the z-axis (D, E, and xy images in C). Cross-sectional z-stack analysis is also shown (xz images in C). Line plots in the lower panels represent the fluorescence signal intensity along the yellow arrows in the respective upper panels. Scale bar, 10 µm. Source data are available online for this figure.

that TMEM25 participates in apico-basal polarization. To test this possibility, we cultured MDCK cells in Matrigel 3D culture and monitored their ability for correct cyst formation. As shown in Fig. 4C, control cells formed a normal cyst with a single lumen, the surface of which was positive for the apical marker protein gp135 (podocalyxin). In contrast, depletion of TMEM25 in MDCK cells led to formation of aberrant cysts, such as those containing one or more cells with inverted polarity (i.e., with gp135 facing the ECM and the basolateral marker β-catenin facing the other side) or those with multiple lumens (Fig. 4C). Thus, TMEM25 appears to contribute to apico-basal polarization during cyst development. Although non-muscle myosin 2 filaments, containing phosphorylated myosin light chain 2 (MLC2), are known to be abundant in the F-actin-rich apicolateral domain of polarized epithelial cells (Quintanilla et al, 2023), phosphorylated MLC2 still colocalized with F-actin in individual cells that constitute aberrant cysts (Fig. EV5A), suggesting that actomyosin organization is not strongly disturbed by depletion of TMEM25.

## TMEM25 interacts with claudins

TJs are composed of at least three distinct types of integral membrane proteins: claudins, occludin, and the Ig family of transmembrane proteins, such as JAM-A (Zihni et al, 2016). To know whether TMEM25 interacts with these TJ proteins, we expressed HA–TMEM25 with claudin-1, occludin, or JAM-A as a FLAG-tagged protein, followed by a co-immunoprecipitation assay. Intriguingly, as shown in Fig. 5A, TMEM25 was capable of strongly interacting with claudin-1. The interaction seems to be specific to claudin, because TMEM25 did not bind to occludin or JAM-A (Fig. 5A). The mammalian claudin family contains 27 members, which are divided into four phylogenetic clusters (Günzel and Fromm, 2012). TMEM25 also interacted with claudin-2: claudin-1 and claudin-2 belong to cluster II. On the other hand, it failed to bind to claudin-4, claudin-15, and claudin-25, which are classified into clusters I, III, and IV, respectively (Fig. 5B). Claudin-1 expressed in HEK293 cells was efficiently co-immunoprecipitated with TMEM25-FL and TMEM25-ΔICR (Fig. 5C, see also Fig. 1A), whereas truncation of the ECR resulted in a complete loss of interaction (Fig. 5C). Thus, TMEM25 likely interacts with claudin-1 via the ECR. *N*-glycosylation in the ECR appears to be dispensable for the interaction, because TMEM25-NQ, an *N*-glycan-deficient mutant, fully bound to claudin-1 (Fig. 5C).

To investigate the mode of TMEM25–claudin-1 interaction, we performed the PLA using HEK293 cells expressing TMEM25–HA and/or FLAG–claudin-1. As shown in Fig. 5D, the PLA signal was

detected on the cell surface solely in a cell expressing both proteins, indicating that TMEM25 interacts with claudin-1 in a *cis*-configuration at the plasma membrane. Furthermore, we tested the possibility that TMEM25 and claudin-1 also associate in a *trans*-manner. For this purpose, FLAG–TMEM25-expressing HEK293 cells and Myc–claudin-1-containing cells were co-cultured, and proteins in the lysate were immunoprecipitated with the anti-FLAG antibody. As shown in Fig. 5E, TMEM25 *trans*-interacted with claudin-1 in an ECR-dependent manner. Thus, TMEM25–claudin interaction likely occurs in both *cis*- and *trans*-configuration.

## TMEM25 suppresses claudin assembly

Oligomer formation of claudins is essential for structural and functional development of TJs (Tsukita et al, 2019; Otani and Furuse, 2020; Piontek et al, 2020). To investigate the effect of TMEM25 on claudin oligomerization, we performed a co-immunoprecipitation assay using HEK293 cells, which do not express endogenous claudins, thus being suitable for study on claudin oligomerization (Piontek et al, 2008; Harris et al, 2010). As shown in Fig. 6A, *cis*-interaction between Myc–claudin-1 and FLAG–claudin-1 was blocked by co-expression of HA–TMEM25. A similar result was obtained when claudin-2 was used instead of claudin-1 (Fig. 6A). TMEM25 is thus capable of preventing claudin *cis*-interaction. Furthermore, expression of TMEM25 resulted in an impaired *trans*-interaction between FLAG–claudin-1 and Myc–claudin-1 (Fig. 6B). On the other hand, *trans*-oligomerization of occludin was not perturbed by expression of TMEM25 (Fig. 6B). Furthermore, JAM-A, another Ig-domain protein at the TJ, failed to affect self-association of claudin-1 in both *cis*- and *trans*-manners (Fig. 6C,D). Thus, TMEM25 appears to specifically attenuate *cis*- and *trans*-oligomerizations of claudins. We further examined the effect of TMEM25 on claudin assembly by analyzing the localization of ectopically expressed claudin-1 and claudin-2 in HEK293 cells, which are free of endogenous claudins 1 to 5 and of TJ strands (Piontek et al, 2008). It is known that in claudin-deficient cells such as HEK293 cells and mouse L cells, ectopic expression of claudins leads to claudin strand formation via *trans*-interaction, as visualized by their accumulation at cell–cell contact regions (Furuse et al, 1998; Piontek et al, 2008; Milatz et al, 2015). As shown in Fig. 6E, FLAG-tagged claudin-1 or claudin-2 was strongly enriched at contact sites between two claudin-expressing cells. The enrichment is considered to reflect claudin *trans*-oligomerization and subsequent strand formation, because it occurred solely when both cells expressed claudin (Fig. 6E).

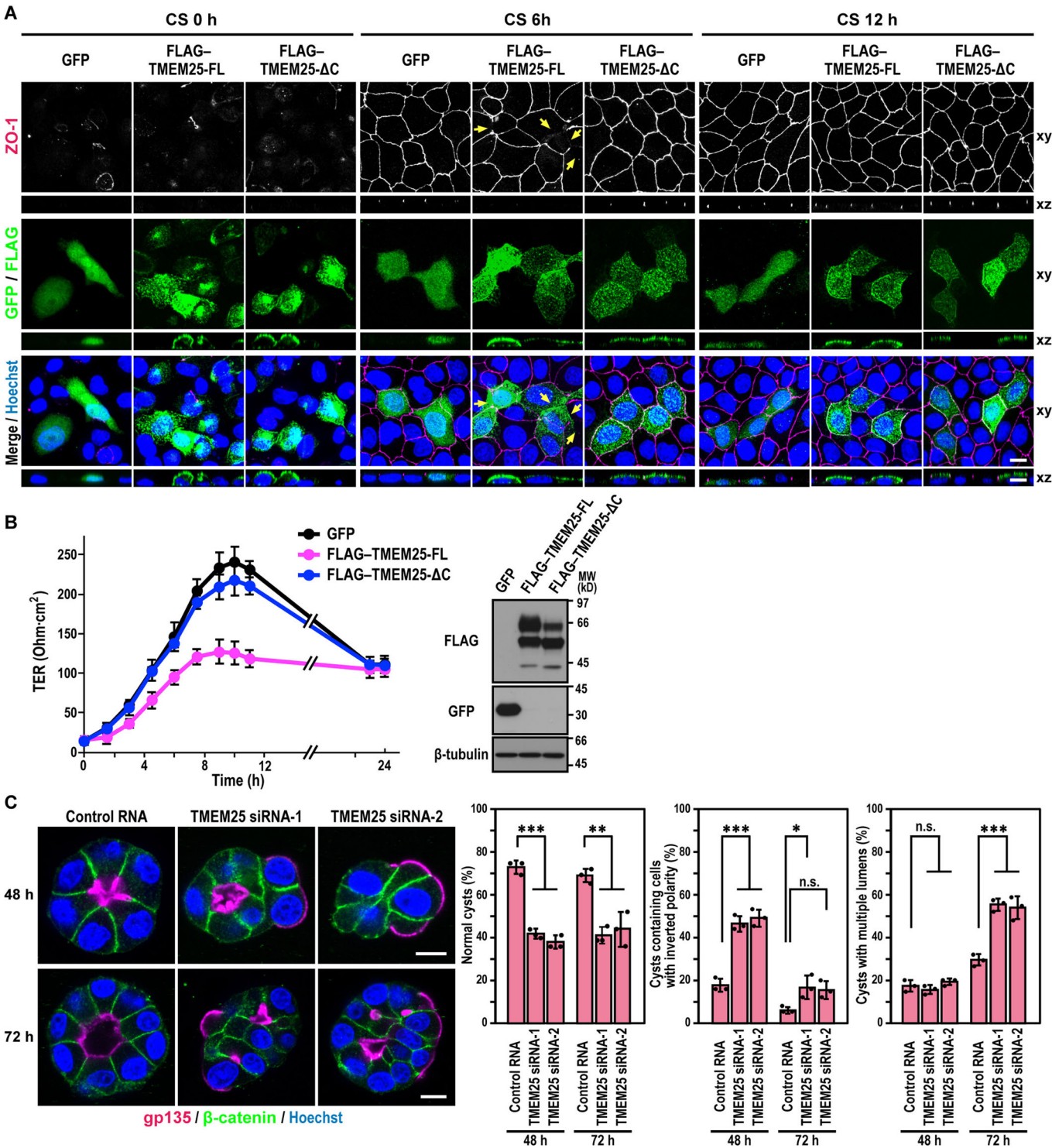

Intriguingly, the claudin assembly at cell–cell contact regions was effectively prevented by co-expression of HA–TMEM25; claudins were not enriched at cell–cell contacts but distributed throughout the plasma membrane in TMEM25-overexpressing cells (Fig. 6E). Furthermore, in MDCK cells, over-expression of TMEM25 impaired accumulation of endogenous claudin-1 and claudin-2 at cell–cell contact regions (Fig. EV5B). These findings indicate that

TMEM25 suppresses claudin assembly and subsequent strand formation.

## Par3 attenuates TMEM25–claudin interaction

To know whether the TMEM25-binding protein Par3 controls TMEM25–claudin interaction, we investigated the effect of

**Figure 4.** **Effect of TMEM25 overexpression on TJ development and cystogenesis of MDCK cells.**

(A) Representative images of MDCK cells expressing GFP alone or FLAG-tagged TMEM25 induced by the Tet-on system in the presence of doxycycline (5 µg/ml). Cells were fixed 0 h, 6 h, or 12 h after $Ca^{2+}$ switch (CS) and stained with the indicated antibodies and Hoechst. Confocal images of the cells were projected along the z-axis (xy). Cross-sectional z-stack analysis is also shown (xz). Arrows indicate the delay in TJ formation in the FLAG–TMEM25-expressing cells. Scale bar, 10 µm. (B) $Ca^{2+}$-induced TER changes of monolayers of MDCK cells expressing FLAG–TMEM25. GFP alone, FLAG–TMEM25-FL, or FLAG–TMEM25-ΔC was expressed in MDCK cells by lentivirus infection. TER of monolayers of the cells was monitored after $Ca^{2+}$ switch (left): values are means ± S.D. from three independent experiments. The expression of GFP or FLAG–TMEM25 proteins were analyzed by immunoblot with the indicated antibodies (right): positions for marker proteins are indicated in kilodaltons (kD). (C) MDCK cells were transfected with control RNA or TMEM25 siRNA and grown for 48 h or 72 h in 3D culture and stained as indicated. Shown are representative confocal images of cysts and quantification of the cyst phenotypes. Values are means ± S.D. from three independent experiments ($n \geq 75$ cysts/ experiment). ***$P < 0.001$; **$P < 0.01$; *$P < 0.05$; and n.s., not significant (Tukey-Kramer test). Scale bar, 10 µm. Source data are available online for this figure.

overexpression of Par3 on TMEM25 binding to claudin-1 in an immunoprecipitation assay using HEK293 cells. As shown in Fig. 6F, the interaction between full-length TMEM25 and claudin-1 was attenuated by co-expression of Par3. Although TMEM25-ΔC, a mutant protein incapable of interacting with Par3 (Fig. 1), was still able to associate with claudin-1, the association was not affected by co-expression of Par3 (Fig. 6F). Thus, Par3 likely have a potential ability to modulate TMEM25–claudin-1 interaction by binding directly to TMEM25, which is consistent with the observation that Par3 interacted with TMEM25 but not with claudin-1 (Fig. 6G). In the process, Par3 does not seem to serve via excluding TMEM25 from the TJ, because its overexpression did not impair TJ localization of TMEM25 under the present experimental conditions (Fig. EV5C); however, it is still possible that Par3 sequesters TMEM25 from claudins within the TJ. Taken together with the present findings, TMEM25 interacts with claudin via its extracellular domain to regulate claudin assembly and TJ formation, and the interaction can be controlled by Par3 binding to the cytoplasmic tail of TMEM25.

## Discussion

In the present study, we show that the Ig-family transmembrane glycoprotein TMEM25 localizes to TJs in epithelial cells and binds to the cell polarity protein Par3 via the interaction between the TMEM25 C-terminus and Par3-PDZ2/3 (Fig. 1). TMEM25 is able to self-associate in *cis* but not in *trans*, suggesting that it is not directly involved in cell–cell contact (Fig. 2). TJ formation in MDCK cells after $Ca^{2+}$ switch is accelerated by depletion of TMEM25 (Fig. 3), and prevented by over-expression of TMEM25 but not by that of TMEM25-ΔC (Fig. 4A,B), indicating that TMEM25 regulates TJ development in conjunction with Par3 during epithelial cell polarization. The significance of TMEM25 in cell polarity control is also demonstrated by an aberrant cyst formation in TMEM25-depleted MDCK cells in 3D culture (Fig. 4C). Intriguingly, TMEM25 specifically interacts with claudin-1 and claudin-2 via the N-terminal extracellular region (ECR) and prevents their assembly (Figs. 5 and 6). TMEM25–claudin interaction is attenuated in a manner dependent on the C-terminal tail of TMEM25 (Fig. 6). Thus, TMEM25 likely regulates claudin assembly to organize TJ formation, which is modulated by its interaction with Par3.

Claudin polymers constitute the backbone of TJ, and the formation of TJ strands requires both *cis*- and *trans*-interactions of claudins (Suzuki et al, 2015; Tsukita et al, 2019; Piontek et al, 2020). However, the molecular mechanism underlying regulation of claudin assembly has remained largely unknown. The present study demonstrates that

the regulation likely involves TMEM25, a novel partner of the polarity protein Par3. TMEM25 seems to be incorporated or associated with the TJ strand, as it accumulates at the TJ to *cis*- and *trans*-interact with claudins (Figs. 1 and 5). In contrast to other TJ proteins such as claudins, occludin, and JAM-A, TMEM25 does not appear to function as a cell–cell adhesion molecule as indicated by its failure to self-associate intercellularly (Fig. 2). Instead, TMEM25 has an ability to prevent both *cis*- and *trans*-interactions of claudin (Fig. 6), which is consistent with its role as a negative regulator of TJ strand formation (Figs. 3 and 4). Since claudin *cis*-oligomerization is assumed to occur at least partially before *trans*-interaction triggers polymerization into TJ strands (Günzel and Fromm, 2012; Gong and Hou, 2017), TMEM25 may primarily inhibit *cis*-association of claudin, leading to the secondary blockade of *trans*-interaction. The anti-parallel double row model of TJ strands, based on the crystal structures of mammalian claudins (Suzuki et al, 2014; Saitoh et al, 2015), shows that the *cis*-interaction of claudins is mediated via the linear *cis*-interface (for formation of a row) and the lateral *cis*-interface (for stabilization of two rows in an antiparallel orientation), both of which are formed by their extracellular segments (Suzuki et al, 2015; Tsukita et al, 2019; Piontek et al, 2020). The significant role of the extracellular segments in claudin *cis*-interaction is consistent with the present finding that TMEM25, a regulator of claudin *cis*-assembly, is capable of *cis*-interacting with claudin via its extracellular domain (Fig. 5C). It still remains possible, however, that TMEM25 prevents claudin *trans*-association in a direct manner as well, because TMEM25 is also able to *trans*-interact with claudin-1 via the ECR (Fig. 5E).

The prevention of claudin assembly by TMEM25 is mediated via its ECR (Fig. 5), whereas Par3 modulates TMEM25–claudin interaction by binding to the C-terminal cytoplasmic tail of TMEM25 (Fig. 6). Although the precise mechanism how Par3 affects TMEM25–claudin interaction is presently unknown, it may involve homo-oligomer formation of Par3, which is mediated via the N-terminal CR1 domain (Mizuno et al, 2003; Feng et al, 2007). Because Par3 interacts with TMEM25 via its PDZ3 domain (Fig. 1), the CR1-mediated self-oligomerization of Par3 may result in a cluster formation of TMEM25, which is possibly involved in regulation of TJ development. The significance of Par3-PDZ3 in promotion of TJ assembly in MDCK cells has been described in a previous study (Chen and Macara, 2005).

Overexpression of Par3 results in an impaired TMEM25–claudin interaction via binding to the TMEM25 C-terminus (Fig. 6). Considering that TMEM25 is able to inhibit claudin–claudin interaction (Fig. 6), the impairment is expected to upregulate claudin oligomerization, raising a possibility that Par3 may play a positive role in claudin assembly. On the other hand, the C-terminal tetrapeptide of TMEM25 is required for TMEM25-induced delay of TJ development in the $Ca^{2+}$-switch assay

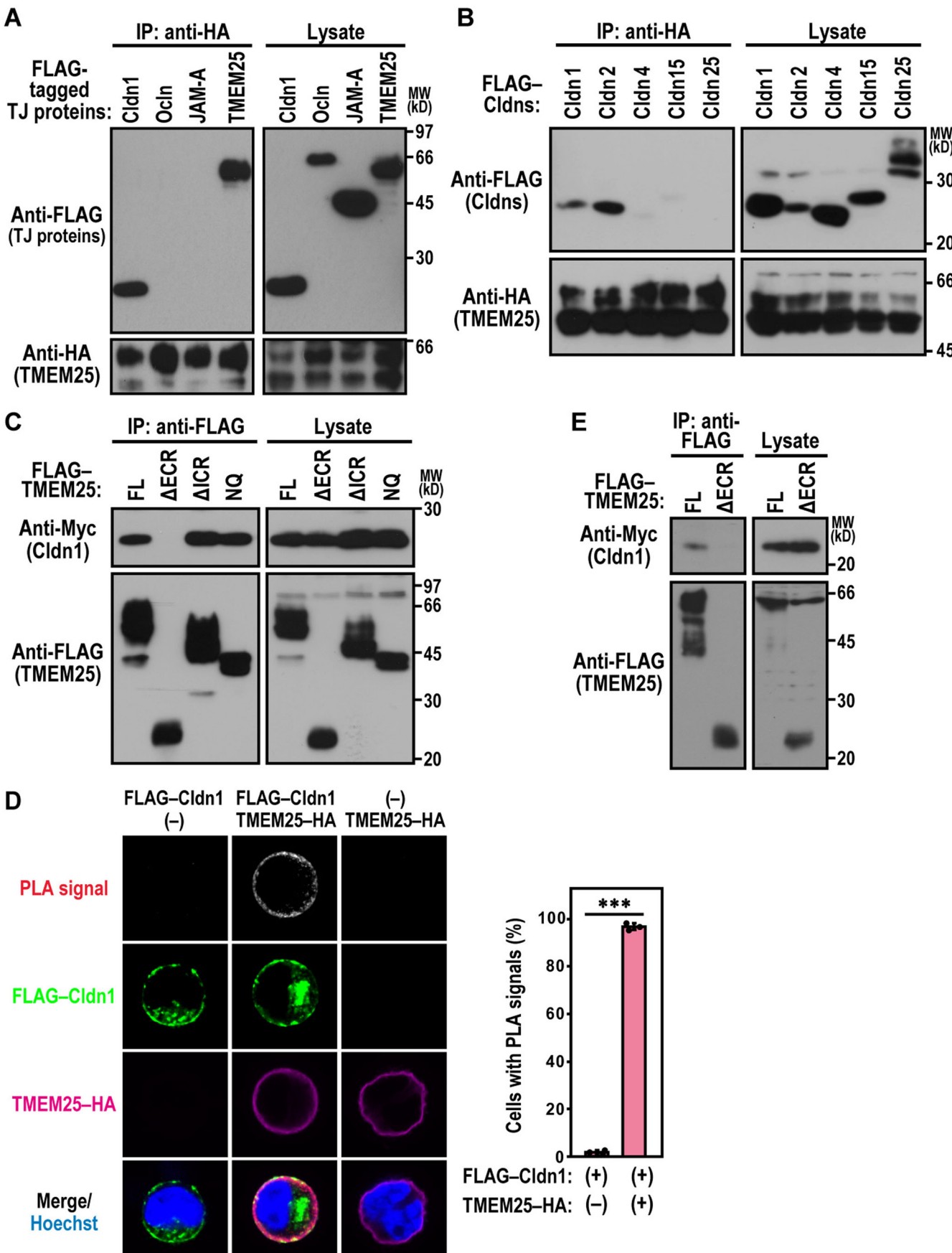

Figure 5.   Interaction of TMEM25 with claudins.

(A–C) *cis*-Interaction of TMEM25 with claudins. HEK293 cells co-expressing HA–TMEM25 with FLAG-tagged claudin-1 (Cldn1), occludin (Ocln), JAM-A, Cldn2, Cldn4, Cldn15, or Cldn25 were cultured at low density (A,B). For determination of regions for *cis*-interaction of TMEM25 with Cldn1, Myc–Cldn1 was co-expressed in HEK293 cells with the indicated form of FLAG–TMEM25 (C). Proteins in the cell lysate were immunoprecipitated (IP) with the anti-HA or anti-FLAG antibody, followed by immunoblot analysis with the indicated antibody. FL, full-length; ECR, extracellular region; ICR, intracellular region; and NQ, a mutation leading to the N106Q/N119Q/N162Q/N175Q/N192Q/N205Q substitution. Positions for marker proteins are indicated in kilodaltons (kD). (D) The PLA for detection of the *cis*-interaction between TMEM25 and claudin-1. A HEK293 cell expressing FLAG–Cldn1 and/or TMEM25–HA was fixed and subjected to the PLA using the anti-FLAG rabbit polyclonal and anti-HA mouse monoclonal antibodies as primary antibodies. Bar graphs indicate quantification of cells with PLA signals at the plasma membrane. Values are means ± S.D. from three independent experiments ($n \geq 110$ cells/ experiment). ***$P < 0.001$ (Student's *t*-test). Scale bar, 10 μm. (E) *trans*-Interaction of TMEM25 with claudin-1. HEK293 cells expressing FLAG–TMEM25-FL or FLAG–TMEM25-ΔECR were co-cultured with Myc–Cldn1-expressing HEK293 cells. Proteins in the lysate of co-cultured cells were immunoprecipitated (IP) with the anti-FLAG antibody, followed by immunoblot analysis with the anti-Myc or anti-FLAG antibody. Source data are available online for this figure.

(Fig. 4A,B), implying a negative regulation of claudin assembly by the C-terminal tail-binding protein Par3. Although the reason for this apparent discrepancy is presently obscure, it seems possible that Par3 at low concentrations does not largely affect the affinity of TMEM25 for claudin but induces a clustering (or a conformational change) of TMEM25, thereby preventing claudin assembly (Figs. 5 and 6); the prevention may contribute at least partially to a delay of TJ formation in MDCK cells with a low expression of Par3 (Fig. 3). In contrast, Par3 at high concentrations may profoundly disrupt TMEM25–claudin interaction (Fig. 6) to release TMEM25-mediated inhibition of claudin assembly, which appears to be consistent with a previous observation that overexpression of Par3 in MDCK cells promotes TJ development in the $Ca^{2+}$-switch assay (Hirose et al, 2002).

It has been shown that the *TMEM25* gene is under-expressed in breast and colorectal cancers (Doolan et al, 2009; Hrašovec et al, 2013), and proposed to be used as a tumor biomarker of favorable prognosis (Doolan et al, 2009). Since TMEM25 is a regulator of claudin assembly as demonstrated in the present study, its low expression probably leads to an aberrant regulation of TJ development and contributes to cancer progression. Indeed, depletion of TMEM25 induces an abnormal cyst formation in 3D culture of MDCK cells, indicative of an impaired cell polarity (Fig. 4C); and TJ structure alteration and subsequent impaired polarization are considered to result in a multitude of diseases, especially adenocarcinoma of various organs (Royer and Lu, 2011; Muthuswamy and Xue, 2012; Garcia et al, 2018; Zeisel et al, 2019). In this context, it should be noted that claudins are considered to participate in cancer metastasis, depending on types of cancers (Martin et al, 2011; Tabariès and Siegel, 2017; Osanai et al, 2017; Bhat et al, 2020): for example, lower expression of claudin-1 is associated with cancer progression and invasion in some cancers, whereas loss of claudin-1 improves the patient survival in other types of the tumors. In addition, Par3, a partner of the claudin assembly regulator TMEM25, is also known to be involved in tumorigenesis and metastasis (Zen et al, 2009; Iden et al, 2012; McCaffrey et al, 2012; Xue et al, 2013; Kitaichi et al, 2017; Jung et al, 2019). Taken together, the TMEM25-containing machinery for claudin assembly regulation could be a potential therapeutic target in cancer management.

## Methods

### Plasmids

The cDNAs encoding human TMEM25, claudin-1, claudin-2, claudin-4, claudin-15, claudin-25, occludin, and JAM-A were prepared by PCR using Human Multiple Tissue cDNA panels (BD Biosciences); the cDNA for human Par3 was cloned as described previously (Kohjima et al, 2002). The cDNA fragments for various regions of Par3 and TMEM25 (summarized in Fig. 1A) were amplified by PCR using specific primers and ligated to the indicated expression vector: Par3-PDZ1 (amino acid residues 221–321), Par3-PDZ2 (residues 408–510), Par3-PDZ3 (residues 525–626), Par3-PDZ2/3 (residues 408–626), TMEM25-262–366 (residues 262–366), and TMEM25-262–362 (residues 262–362) to pGADGH or pGBK, a modified pGBT vector, for yeast two-hybrid experiments; TMEM25-ECR (residues 27–220), TMEM25-262–366 (residues 262–366), TMEM25-262–362 (residues 262–362), TMEM25-355–366 (residues 355–366), and TMEM25-355–362 (residues 355–362) to pGEX-6P (GE Healthcare Biosciences) for expression as glutathione *S*-transferase (GST)-fusion protein in *Escherichia coli*; TMEM25-ECR (residues 27–220), Par3-PDZ1 (residues 221–321), Par3-PDZ2 (residues 408–510), Par3-PDZ3 (residues 525–626), and Par3-PDZ2/3 (residues 408–626) to pMAL-c2 (New England Biolabs) for expression as protein fused to maltose-binding protein (MBP) in *Escherichia coli*; the full-length (FL) of Par3 (residues 1–1,244), claudin-1 (residues 1–211), claudin-2 (residues 1–230), claudin-4 (residues 1–209), claudin-15 (residues 1–228), claudin-25 (residues 1–253), and occludin (residues 1–522) to pEF-BOS or pcDNA3 (Thermo Fisher Scientific) for expression in mammalian cells; and TMEM25-FL (residues 27–366), TMEM25-ΔC (residues 27–362), TMEM25-ΔECR (residues 225–366), TMEM25-ΔICR (residues 27–268), JAM-A (residues 28–299), and TM-JAM chimera, a mutant TMEM25 protein with the replacement of its transmembrane segment (residues 238–260) with the corresponding region of JAM-A (residues 238–261), to pCMV-8 (Sigma-Aldrich), pTetOne (Clontech), or pLVSIN-CMV-Neo (Takara Bio), which are modified to contain the signal peptide sequence of preprotrypsin preceding the FLAG-, HA-, or Myc-tags, for expression of N-terminally tagged type I transmembrane proteins. For the PLA of TMEM25–claudin-1 interaction, pCMV-8 was modified for expression of C-terminally HA-tagged TMEM25-FL (residues 27–366) in mammalian cells. All of the constructs were sequenced for confirmation of their identities.

### Antibodies

Anti-FLAG rabbit polyclonal antibodies and the anti-FLAG (M2) and anti-β-tubulin (TUB 2.1) mouse monoclonal antibodies were purchased from Sigma-Aldrich; the anti-FLAG mouse monoclonal antibody (1E6) from FUJIFILM Wako Pure Chemical Corporation;

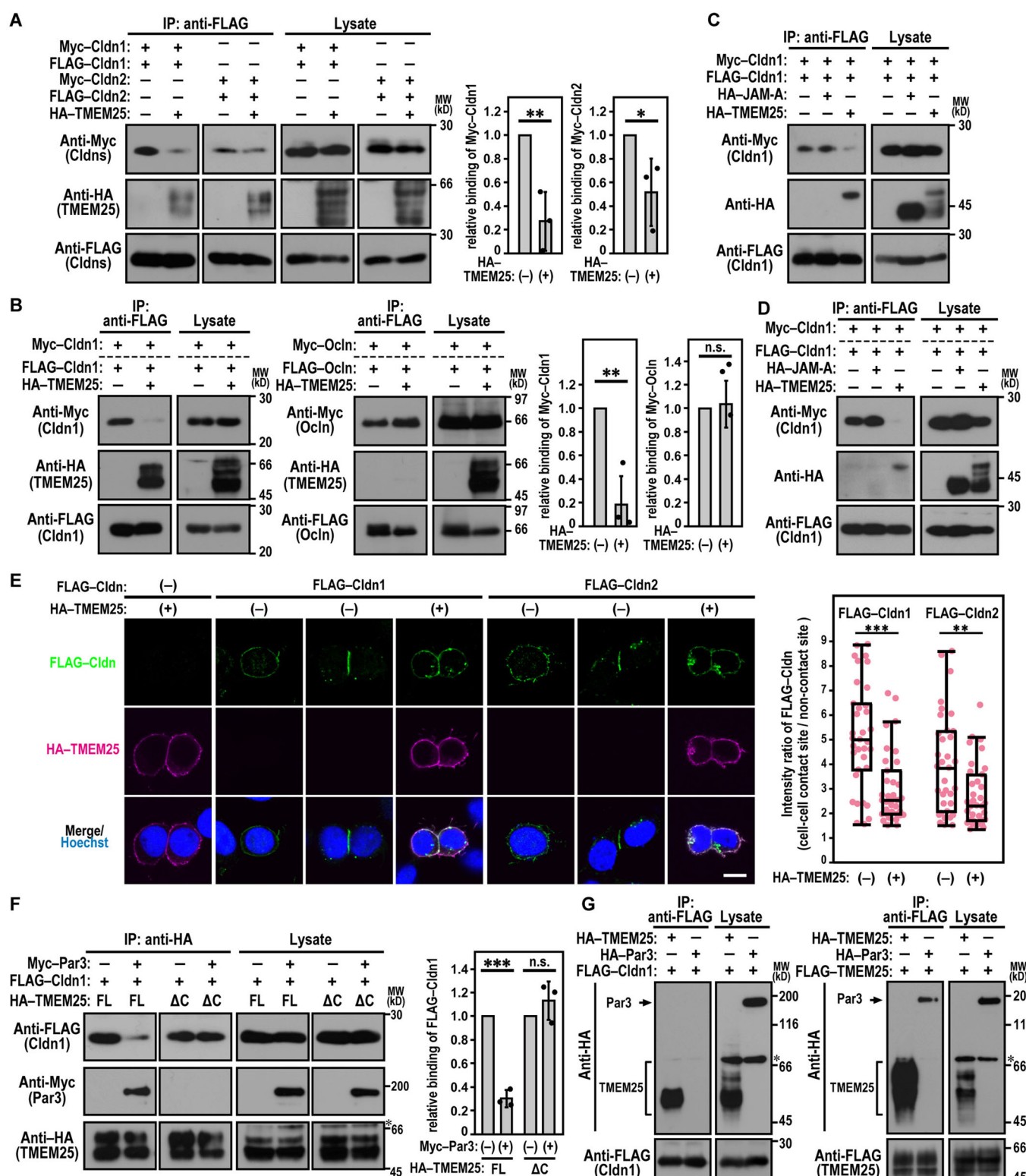

the anti-HA mouse monoclonal antibody (16B12) from Covance; the anti-ZO-1 rat monoclonal antibody (R40.76), anti-β-catenin (H-102), anti-HA (Y-11), and anti-Myc (A14) rabbit polyclonal antibodies from Santa Cruz Biotechnology; the anti-Myc (9E10) mouse monoclonal and anti-HA (3F10) rat monoclonal antibodies

from Roche Applied Science; anti-MBP rabbit polyclonal antibodies from New England Biolabs; anti-Par3 rabbit polyclonal antibodies from EMD Millipore; the anti-occludin mouse monoclonal antibody (OC-3F10), anti-ZO-1, anti-claudin-1, and anti-claudin-2 rabbit polyclonal antibodies, and Alexa Fluor 647-conjugated anti-

**Figure 6. TMEM25-mediated suppression of claudin assembly and Par3-induced attenuation of TMEM25–claudin interaction.**

(A) Suppression of claudin *cis*-interaction by TMEM25. HEK293 cells co-expressing FLAG-Cldn1/2, Myc-Cldn1/2 and HA-TMEM25 were cultured at low density, and proteins in the cell lysate were immunoprecipitated (IP) with the anti-FLAG antibody, followed by immunoblot analysis with the indicated antibody. Proteins in the cell lysate were immunoprecipitated (IP) with the anti-FLAG antibody, followed by immunoblot analysis with the indicated antibody. Positions for marker proteins are indicated in kilodaltons (kD). Bar graphs indicate the relative ratios of Myc–Cldn to FLAG–Cldn in the precipitates; the ratio of Myc–Cldn to FLAG–Cldn in the absence (−) of HA–TMEM25 is set as 1.0. Values are means ± S.D. from three independent experiments. **$P < 0.01$ and *$P < 0.05$ (Student's *t*-test). Cldn, claudin. (B) Suppression of claudin *trans*-interaction by TMEM25. HEK293 cells containing FLAG-Cldn1 with or without HA-TMEM25 were co-cultured with Myc-Cldn1-expressing HEK293 cells to confluency (left panel); or HEK293 cells containing FLAG-Ocln with or without HA-TMEM25 were co-cultured with Myc-Ocln-expressing HEK293 cells to confluency (right panel). Proteins in the lysate of co-cultured cells were immunoprecipitated (IP) with the anti-FLAG antibody, followed by immunoblot analysis with the indicated antibody. Bar graphs indicate the relative ratios of Myc–Cldn1 to FLAG–Cldn1 or Myc–Ocln to FLAG–Ocln in the precipitates; the ratio of the Myc-tagged protein to the FLAG-tagged protein in the absence (−) of HA–TMEM25 is set as 1.0. Values are means ± S.D. from three independent experiments. **$P < 0.01$ and n.s., not significant (Student's *t*-test). Ocln, occludin. (C) Effect of JAM-A expression on claudin *cis*-interaction. HEK293 cells co-expressing FLAG-Cldn1 and Myc-Cldn1 with HA-JAM-A or HA-TMEM25 were cultured at low density, and proteins in the cell lysate were immunoprecipitated (IP) with the anti-FLAG antibody, followed by immunoblot analysis with the indicated antibody. (D) Effect of JAM-A expression on claudin *trans*-interaction. HEK293 cells containing FLAG-Cldn1 with HA-JAM-A or HA-TMEM25 were co-cultured with Myc-Cldn1-expressing HEK293 cells to confluency. Proteins in the cell lysate were immunoprecipitated (IP) with the anti-FLAG antibody, followed by immunoblot analysis with the indicated antibody. (E) TMEM25-mediated suppression of claudin enrichment at contacts between two claudin-expressing cells. HEK293 cells expressing FLAG-Cldn1/2 with (+) or without (−) HA-TMEM25 were fixed and stained as indicated (left). Scatter diagrams and box-and-whisker plots of the fluorescence intensity ratio of FLAG-Cldn1 or FLAG-Cldn2 at the cell–cell contact site to that at the non-contact surface ($n = 35$ cells/ experiment). Boxes represent lower/upper quartiles with the median values indicated with a horizontal line. Whiskers show minimum and maximum values. ***$P < 0.001$ and **$P < 0.01$ (Wilcoxon rank sum test). Scale bar, 10 µm. (F) Attenuation of TMEM25–claudin interaction by Par3. Myc-Par3 were co-expressed with FLAG-Cldn1 and HA-TMEM25-FL or HA-TMEM25-ΔC in HEK293 cells. Proteins in the cell lysate were immunoprecipitated (IP) with the anti-HA antibody, followed by immunoblot analysis with the indicated antibodies. ngth; and ΔC, a mutant protein with deletion of the C-terminal tetrapeptide. Bar graphs indicate the relative ratios of FLAG-Cldn1 to HA-TMEM25 in the precipitates; the ratio of FLAG-Cldn1 to HA-TMEM25 in the absence (−) of Myc-Par3 is set as 1.0. Values are means ± S.D. from three independent experiments. ***$P < 0.001$ and n.s., not significant (Student's *t*-test). The asterisk indicates a nonspecific band present in the cell lysates. FL, full-length. (G) Failure of Par3 to interact with claudin-1. FLAG-Cldn1 (left panel) or FLAG-TMEM25 (right panel) was co-expressed with HA-Par3 or HA-TMEM25 in HEK293 cells, and proteins in the cell lysate were immunoprecipitated (IP) with the anti-FLAG antibody, followed by immunoblot analysis with the indicated antibody. The asterisk indicates a nonspecific band present in the cell lysates. Source data are available online for this figure.

ZO-1 mouse monoclonal antibody (ZO-1A12) from Thermo Fisher Scientific; the anti-E-cadherin rabbit monoclonal antibody (24E10) and anti-phospho-myosin light chain 2 (MLC2) (Thr18/Ser19) rabbit polyclonal antibodies from Cell Signaling Technology; anti-TMEM25 rabbit polyclonal antibodies (A13805) from Boster; and the anti-GFP rat monoclonal (GF090R) or mouse monoclonal (GF200) antibodies from Nacalai Tesque.

## Two-hybrid experiments

The yeast two-hybrid screening was performed using a human fetal brain cDNA library (Clontech) in the reporter strain AH109 using the second and third PDZ domains of human Par3 (amino acid residues 408–626) as a bait, as previously described (Izaki et al, 2005; Iwakiri et al, 2013). Among $1.8 \times 10^6$ clones screened, a positive clone obtained encodes the intracellular region of TMEM25 (amino acid residues 276–366) as revealed by sequencing analysis. For analysis of the interaction between TMEM25 and Par3, HF7c cells were transformed with combinations of pGADGH encoding TMEM25 and pGBK encoding Par3. After selection for the Trp+ and Leu+ phenotypes, transformed cells were tested for their ability to grow on plates lacking histidine supplemented with 5 mM 3-aminotriazole.

## Cell culture

Madin-Darby canine kidney (MDCK) II cells and human embryonic kidney HEK293T cells were cultured in Dulbecco's modified Eagle's medium (DMEM) with 10% FCS. Caco-2 cells were cultured in Eagle's minimal essential medium (MEM) with 20% FCS.

## GST pull-down assay

GST- or MBP-tagged proteins were prepared as previously described (Hayase et al, 2013). Briefly, recombinant proteins were expressed in

the *Escherichia coli* strain BL21, and cells were homogenized by sonication; after centrifugation, GST- or MBP-tagged proteins were purified using glutathione-Sepharose 4B (GE Healthcare Biosciences) or amylose resin (New England Biolabs), respectively. GST–TMEM25-(355–366) or GST–TMEM25-(355–362) was incubated with MBP-tagged PDZ domains of Par3 for 20 min at 4 ˚C in 500 µl of a solution containing 100 mM NaCl, 1 mM dithiothreitol (DTT), 0.1% Triton X-100, and 20 mM Tris-HCl, pH7.6. GST- and MBP-tagged TMEM25-ECR proteins were incubated for 1 h at 4 ˚C in 500 µl of a solution containing 150 mM NaCl, 0.1% Triton X-100, and 20 mM Tris-HCl, pH7.6. Proteins pulled down with glutathione-Sepharose beads were subjected to SDS-PAGE, followed by staining with *Coomassie Brilliant Blue* (CBB) or immunoblot analysis with the anti-MBP polyclonal antibodies.

## Co-immunoprecipitation analysis

HEK293T cells were transfected with the following plasmid using X-tremeGENE™ HP (Roche Applied Science): pCMV-8 for TMEM25 or JAM-A as FLAG- or HA-tagged protein; pEF-BOS for claudin-1, -2, -4, -15, or -25 as FLAG- or Myc-tagged protein; or pcDNA3 for Par3 or occludin as FLAG-, HA-, or Myc-tagged protein.

Analysis of *cis*-interaction between proteins was performed according to the method of Hou et al (Hou et al, 2010) with minor modifications. Briefly, HEK293T cells were co-transfected with two or three plasmids, each encoding the indicated protein with a FLAG-, HA-, or Myc-tag. Doubly or triply transfected cells were cultured at low density to minimize cell–cell contacts and *trans*-interactions. The cells co-expressing a FLAG-tagged protein with an HA- and/or Myc-tagged protein were lysed with a lysis buffer (150 mM NaCl, 4 mM EDTA, 10% glycerol, 0.5% Triton X-100, and 50 mM Tris-HCl, pH7.5) containing 1 mM DTT and Protease Inhibitor Cocktail (Sigma-Aldrich).

Analysis of *trans*-interaction between proteins was performed according to the method of Daugherty et al (Daugherty et al, 2007) with minor modifications. Briefly, HEK293T cells expressing the following different protein were co-cultured to confluency: HEK293T expressing HA–TMEM25 and HEK293T expressing FLAG–TMEM25 (Fig. 2G); HEK293T expressing HA–JAM-A and HEK293T expressing FLAG–JAM-A (Fig. 2G); HEK293T expressing Myc–Cldn1 and HEK293T expressing FLAG–Cldn1 with or without HA–TMEM25 (Fig. 6B); HEK293T expressing Myc–Ocln and HEK293T expressing FLAG–Ocln with or without HA–TMEM25 (Fig. 6B); and HEK293T expressing Myc–Cldn1 and HEK293T expressing FLAG–Cldn1 and HA-tagged TMEM25 or JAM-A (Fig. 6D). The co-cultured cells were lysed with the lysis buffer containing Protease Inhibitor Cocktail.

The cell lysates were precipitated with the anti-FLAG (M2) or anti-HA (3F10) monoclonal antibody, coupled to protein G-Sepharose (GE Healthcare) as previously described (Kamakura et al, 2013). The precipitates were washed twice with the lysis buffer containing 1 mM DTT and applied to SDS-PAGE, followed by immunoblot analysis using the anti-FLAG, HA, or Myc polyclonal antibodies. The blots were developed using ImmunoStar LD (FUJIFILM Wako Pure Chemical Corporation) or ImmunoStar Zeta (FUJIFILM Wako Pure Chemical Corporation) for visualization of the proteins. Band intensities in the precipitate and in the lysates were measured using ImageJ software (version 1.53k, National Institutes of Health). Within each experiment, the intensity of the band in the precipitate was normalized by the corresponding value in the lysates and expressed as a ratio as indicated in figure legends.

## Glycosidase treatment

MDCKII cells expressing FLAG–TMEM25 were lysed with the lysis buffer containing 1 mM DTT and Protease Inhibitor Cocktail. The lysates were centrifuged for 20 min at $20,000 \times g$, and the resultant supernatant was subjected to treatment with PNGase-F (New England Biolabs) or Endo-H (New England Biolabs). Proteins were applied to 10% SDS-PAGE, followed by immunoblot analysis with the anti-FLAG antibody.

## Knockdown with siRNA

As double strand small interfering RNAs targeting TMEM25 and Par3, the sequences on the sense strand of 25-nucleotide modified synthetic RNAs (Stealth™ RNAi; Thermo Fisher Scientific) used were as follows: TMEM25-siRNA-1, 5′-UCUCUGAUCUCUAGUGACUCCAACA-3′; TMEM25-siRNA-2, 5′-CAGCCAACGCUUCUGUCAUCCUCAA-3′; TMEM25-siRNA-3, 5′-ACGCCUCUGUCAUCCUUAAUGUGCA-3′; TMEM25-siRNA-4, 5′- UCUCUGAUAUCAAGUGACUCCAACA-3′; Par3-siRNA-1, 5′-GGCUUCGGGUGAAUGAUCAACUGAU-3′; and Par3-siRNA-2, 5′-CCAUGUGGUUCCUGCAGCAAAUAAA-3′. Medium GC Duplex of Stealth™ RNAi Negative Control Duplexes #2 (Thermo Fisher Scientific) was used as a negative control. MDCKII cells plated at $2.7 \times 10^4/cm^2$ were transfected with siRNA (6.4 pmol) using LipofectAMINE™ RNAiMAX (Thermo Fisher Scientific) and cultured for 48–72 h in DMEM containing 10% FCS.

## Calcium switch assay

The calcium switch assay was performed as previously described (Hayase et al, 2013; Chishiki et al, 2017). Briefly, MDCKII cells

$(7.5 \times 10^5)$ were seeded on a 35 mm glass bottom dish (glass surface diameter 14 mm; Matsunami Glass) and grown in DMEM with 10% FCS containing 1.8 mM calcium (the normal calcium medium), for the formation of confluent monolayers. The monolayer cells were cultured for 24 h in the low calcium medium containing 2.1 µM calcium (S-MEM; Thermo Fisher Scientific) supplemented with dialyzed 5% FCS. The calcium switch assay was initiated by incubation of the cells in the normal calcium medium. For measurement of transepithelial electrical resistance (TER) (Matter and Balda, 2003; Hayase et al, 2013; Chishiki et al, 2017), MDCKII cells $(5 \times 10^6)$ were seeded on a 12 mm Transwell chamber (0.4 µm pore size; Corning Costar). After the addition of calcium, the change in TER was monitored with a Millicel-ERS (EMD Millipore). TER values were calculated by subtracting the blank value from an empty filter and were expressed in ohm•cm². For paracellular flux measurement (Otani et al, 2019; Ikenouchi et al, 2005), MDCKII cells $(5 \times 10^6)$ were seeded on the 12 mm Transwell chamber. After calcium depletion for 24 h, the medium was changed to phenol red-free DMEM (Nacalai Tesque) supplemented with 10% FCS and cells were incubated for the indicated period. Fluorescein-isothiocyanate (FITC)-conjugated dextran of 4 kDa (Sigma-Aldrich) was added to the apical compartment of the Transwell chamber at a concentration of 1 mg/ml and incubated for 30 min. From the basal compartment of the chamber, the medium was collected and the fluorescence intensity was measured with a HITACHI F-2500 fluorescence spectrophotometer with an excitation wavelength of 490 nm and detection of emissions at 520 nm. The apparent permeability (Papp) was determined by the following formula: Papp (cm/s) = {(dQ/dt) × Vacc}/(A × C), where dQ is the amount of tracer transported to the acceptor chamber (determined from standard curves) during time interval dt, Vacc is the acceptor volume, A is the area, and C is the initial tracer concentration.

## Immunofluorescence microscopy

Immunofluorescence microscopy was performed as previously described (Kamakura et al, 2013; Hayase et al, 2013). For staining of TMEM25, ZO-1, claudins, and occludin, MDCKII cells in monolayer (2D) culture were fixed for 15 min in 5% acetic acid in ethanol at −20 °C, followed by blocking for 30 min with 0.5% Triton X-100 in phosphate-buffered saline (PBS; 137 mM NaCl, 2.7 mM KCl, 8.1 mM $Na_2HPO_4$, and 1.5 mM $KH_2PO_4$, pH7.4) containing 3% bovine serum albumin (BSA). The 3D culture of MDCKII cells was performed as previously described (Hayase et al, 2013; Chishiki et al, 2017); briefly, cells were trypsinized to a single cell suspension of $1.6 \times 10^4$ cells/ml in 5% Matrigel, containing laminin, type IV collagen, and entactin (Corning Costar), and 250 µl of the suspension was plated in an 8-well cover glass chamber (Iwaki) precoated with 40 µl of Matrigel. For staining of gp135 and β-catenin, MDCKII cells plated in Matrigel were fixed for 30 min in 4% paraformaldehyde, and subsequently permeabilized for 1 h in PBS containing 0.5% Triton X-100 and 3% BSA. Indirect immunofluorescence analysis of 2D- or 3D-cultured cells was performed using the following as secondary antibody: Alexa Fluor 488-labeled goat anti-rabbit or anti-mouse IgG antibodies, Alexa Fluor 594-labeled goat anti-rat IgG antibodies, or Alexa Fluor 633-labeled goat anti-rat IgG antibodies (Thermo Fisher Scientific). Nuclei were stained with Hoechst 33342 (Thermo Fisher Scientific). Confocal images were captured at room temperature on the

confocal microscope LSM780 (Carl Zeiss) and analyzed using ZEN (Carl Zeiss, Inc.). The microscopes were equipped with a C-Apochromat 63×/1.2 NA W Corr water-immersion objective lens.

## Inducible expression of TMEM25 in MDCK cells

For inducible expression of TMEM25-FL and TMEM25-ΔC, we used the plasmid-based Tet-on system (Tet-One™ Inducible Expression System, Clontech), according to the manufacturer's instructions. MDCKII cells were transfected with pTetOne-FLAG–TMEM25 plasmid by Nucleofector (Lonza), and cultured for 24 h in DMEM with 10% FCS. The cells ($2.5 \times 10^5$) were then plated on a 35 mm glass bottom dish (Matsunami Glass), followed by the addition of doxycycline (Dox; LKT laboratories) 3 h after plating. For analysis of the localization of FLAG-tagged TMEM25, cells were cultured for further 48 h before fixation in the presence of Dox (1 μg/ml). For analysis of TJ formation, the above-mentioned calcium switch assay was performed in the presence of Dox (5 μg/ml).

## Lentiviral transfection

The lentivirus for expression of FLAG–TMEM25 was generated using pLVSIN-CMV-Neo (Takara Bio) and Lentiviral High Titer Packaging Mix (Takara Bio), according to the manufacture's instruction. Briefly, HEK293T cells were cotransfected with pLVSIN-CMV-Neo-FLAG–TMEM25 and Lentiviral High Titer Packaging Mix plasmids by using X-tremeGENE HP DNA Transfection Reagent (Sigma–Aldrich). After culture for 72 h, supernatants from transfected HEK293T cells were collected and filtered with a 0.45 μm filter (Millipore). The viral supernatants were added to subconfluent MDCKII cells, and the infected cells were cultured for 24 h. After the virus-containing medium was replaced with fresh DMEM with 10% FCS, the cells were cultured for another 24 h and then seeded on the Transwell chamber for measurement of TER.

## Proximity ligation assay

The proximity ligation assay (PLA) (Söderberg et al, 2006) was performed using the Duolink® PLA kit (Sigma Aldrich), according to the manufacturer's protocol. Briefly, HEK293 cells expressing FLAG- and HA-tagged proteins were fixed for 10 min in 4% paraformaldehyde, permeabilized for 20 min in PBS containing 0.5% Triton X-100 and 3% BSA, and incubated for 18–24 h at 4 °C with the anti-FLAG rabbit polyclonal (Sigma-Aldrich) and anti-HA mouse monoclonal (Covance) antibodies. Cells were then incubated for 1 h at 37 °C in a humidified chamber with the PLA probes: Duolink® In Situ PLA® Probe Anti-Mouse MINUS (Sigma Aldrich) and Duolink® In Situ PLA® Probe Anti-Rabbit PLUS (Sigma Aldrich). Using Duolink® In Situ Detection Reagents Orange (Sigma Aldrich), ligation reaction was performed for 30 min at 37 °C, followed by amplification for 100 min at 37 °C. For visualization of the primary antibodies, cells were incubated for 45 min at 25 °C with Alexa Fluor 488-labeled goat anti-rabbit (Thermo Fisher Scientific) and Alexa Fluor 647-labeled goat anti-mouse IgG antibodies (Thermo Fisher Scientific).

## Statistical analysis

Statistical differences of data for cystogenesis by siRNA-transfected cells were determined by one-way ANOVA with Tukey-Kramer's multiple comparison of means test. Data obtained by the PLA and the co-immunoprecipitation binding assay were analyzed by Student's t test. Data for the intensity ratio of claudins were analyzed by two-sided Wilcoxon rank sum test.

# Data availability

This study includes no data deposited in external repositories.

# Peer review information

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

## Acknowledgements

We thank Namiko Kubo (Kyushu University) and Natsuko Morinaga (Kyushu University) for technical assistance and Hiromi Takeyama (Kyushu University) for secretarial assistance. We are grateful to technical staffs from The Research Support Center, Research Center for Human Disease Modeling, Kyushu University Graduate School of Medical Sciences. This work was supported in part by JSPS (Japan Society for the Promotion of Science) KAKENHI Grants JP21H02698 (to H.S.), JP21H05267 (to H.S.), and JP22K06901 (to S.K.).

## Author contributions

**Sachiko Kamakura**: Conceptualization; Resources; Data curation; Formal analysis; Supervision; Funding acquisition; Validation; Investigation; Visualization; Methodology; Writing—original draft; Project administration; Writing—review and editing. **Junya Hayase**: Resources; Formal analysis; Investigation; Methodology; Writing—review and editing. **Akira Kohda**: Resources; Data curation; Formal analysis; Investigation; Methodology; Writing—review and editing. **Yuko Iwakiri**: Investigation. **Kanako Chishiki**: Investigation. **Tomoko Izaki**: Investigation. **Hideki Sumimoto**: Conceptualization; Data curation; Formal analysis; Supervision; Funding acquisition; Validation; Writing—original draft; Project administration; Writing—review and editing.

## Disclosure and competing interests statement

The authors declare no competing interests.

# Expanded View Figures

**Figure EV1.   Multiple sequence alignment of TMEM25 proteins.**

The amino acid sequences of TMEM25-related proteins from various species are aligned using the MAFFT program. Positions of the signal sequence, a C2-type immunoglobulin (Ig) domain, a region of an Ig-like fold, a transmembrane segment, and the PSD-95/DLG/ZO-1 (PDZ)-binding motif are indicated above the sequences. Potential *N*-glycosylation sites (Asn-*X*aa-Ser/Thr, where *X*aa is any amino acid but not proline) are underlined in magenta. Hs, *Homo sapiens* (human); Mm, *Mus musculus* (mouse); Cf, *Canis familiaris* (dog); Xt, *Xenopus tropicalis* (frog); Tr, *Takifugu rubripes* (puffy fish); and Ol, *Oryzias latipes* (Medaka fish).

# TMEM25

```
                            signal sequence                       C2-type Ig domain
Human(Hs)      1 M-ALPPGPAALRHTLLLLLPALLSSGWGELEP--QIDG-QTWAERALRENERHAFTCRVAG
Mouse(Mm)      1 M-ELPLSQATLRHTLLLLPALLSSGQGELAP--QIDG-QTWAERALRENEHHAFTCRVAG
Dog(Cf)        1 M-APPPGPAALPRSLLLLLPALLSSGWGELAP--QIDG-QTWAERALRENEQHAFTCRVAG
Frog(Xt)       1 M--------LLRVSLLFFQPLFQRGLG----------------EPLQKDETEGASCEGGG
Fugu(Tr)       1 MGHVCVRSWASSATVMLFHTLTFSWTGAMEPTSKQEGRQQQAAMALQEDVTHQFSCHSDG
Medaka(Ol)     1 MRTASPGSWTSGSAVVFFNTLALSWTGAADSVLKING-QHQAAVTLQENMTLKFNCQTDS
                 *           :::::  .*                *        .*:::     .*.   .

Human(Hs)     57 --GPGTPRLAWYLDGQLQEASTSRLLSVGGEA---------FSGGTSTFTVTAHRAQHEL
Mouse(Mm)     57 --GSATPRLAWYLDGQLQEATTSRLLSVGGDA---------FSGGTSTFTVTAQRSQHEL
Dog(Cf)       57 --GSGTPRLAWYLDGQLQEAGTSRLLSVGGEA---------FSGGTSTFTVTAQRAQHEL
Frog(Xt)      37 ------PSLSWYMNGVKQEEGLGREPPFLLPV---------YPGSSSVLTLVETRGE---
Fugu(Tr)      61 RDPRHPLVIRWHLDGNWQKQEPSKHRRLAMTSGRDSDAVHLGYGHNSTFSLRPRKWNREL
Medaka(Ol)    60 WDPRAPPLLTWYLNGVQQKEPSSNRGRLTATSKKDSKVTRPGTNHNSTFSLQARKWDREL
                   :  *:::*  *:    ..   .           .  .*.:::    :  :

                                                Ig-like region
Human(Hs)    106 NCSLQDPRSGRSANASVILNVQFKPEIAQVGAKYQEAQGPGLLVVLFALVRANPPANVTW
Mouse(Mm)    106 NCSLQDPGSGRPANASVILNVQFKPEIAQVGAKYQEAQGPGLLVVLFALVRANPPANVTW
Dog(Cf)      106 NCSLQDPGSGRSANASVILNVQFKPEIAQVGAKYQDSQGPGLLVVLFALVRANPPANVTW
Frog(Xt)      79 NCS------DAWLKGEELLSVHFPPD-----SPVNSAHTPGISLLLLLVVRTQPASTFTL
Fugu(Tr)     121 VCVASNPRTGERYNATITLSLQFKPEILRVNVNHSETSDPAFALVLFALVRSNPPATISF
Medaka(Ol)   120 VCVALNPSTGQSYNATITLNVQFQPEILRVNAHFTETSDPGLSLVLFALVRSNPSATISF
                 *       :.    :.     *.::* *:         .:  *.: ::*: :**::*.:.:

Human(Hs)    166 IDQDGPVTVNTSDFLVLDAQNYPWLTNHTVQLQLRSLAHNLSVVATNDVGVTSASLPAPG
Mouse(Mm)    166 IDQDGPVTVNASDFLVLDAQNYPWLTNHTVQLQLRSLAHNLSVVATNDVGVTSASLPAPG
Dog(Cf)      166 IDQDGPVTVNTSDFLVLDAQNYPWLTNHTVQLQLRSLAHNLSVVATNDVGVTSASLPAPG
Frog(Xt)     128 RDHDGRKTLNSSRLLLLLDTRNLD--TNGSLRVKVS----------TEERGVSHTSVSALG
Fugu(Tr)     181 VDQSGQLVADTTDFLLLDSQTNPQLANNTLRIMLSSLSGTLSLNVTNTAGTVQSNLTLAE
Medaka(Ol)   180 VDQLGQPVANTSDLLTLDSQRYPWLNNHTLRVRLSSLSGNISLNASNSVGAVQSNLTLAE
                 *: *   .  :::  :*  **::      * ::::  :       ::  *.  :.:.

                                         transmembrane
Human(Hs)    226 LLATRVEVPLLGIVVAAGLALGTLVGFSTLVACLVCR-KEKKTKGPSRHPSLISSDSNNL
Mouse(Mm)    226 LLATRIEVPLLGIVVAGGLALGTLVGFSTLVACLVCR-KEKKTKGPSRRPSLISSDSNNL
Dog(Cf)      226 LLATRVEVPLLGIIMAGGLALGALVGFSTLVACLVCR-KEKKTKGPSRRPSLISSDSNNL
Frog(Xt)     176 LLSSHVEVPLFALVVGGAGVVAGILLVNALVCCLLLKRKRRRSYGVRNQLTL---STSNNM
Fugu(Tr)     242 FLQSRVEVPMLGIVTGGAMAFMALLILSLIVLCLMQKNKSKSFDQPVEIVMTKKSDSASM
Medaka(Ol)   240 FLQSRVEVPMLGIVTGGAMAFMALLILSLIVLCLMQKNKSKSFDEPVEIVMTTKSESANL
                 :*  ::::***::.:: ...  .. ::  .:* **: : * :.       . * * .:

Human(Hs)    285 K---LNNVRLPRENMSLPSNLQLNDLTPDSRAVKPADRQMAQNNSRPELLDPEPGGL---
Mouse(Mm)    285 K---LNNVRLPRENMSLPSNLQLNDLTPDLRG-KATERPMAQHSSRPELLEAEPGGL---
Dog(Cf)      285 K---LNNVRLPRENMSLPSNLQLNDLTPDSRG-KPADRQMAQNNSRPELLDPEPGGL---
Frog(Xt)     233 K---LNNSCLPREHMSLPSNLQLNDLRPQARG-------PLGSS--EGETQEDASLRCG
Fugu(Tr)     301 RAEGAGTSHLPRDHMSLPSHVQLNDLSTLTKA-AQ---QNPGGGKREEEEEEEDLSLV--
Medaka(Ol)   300 QTGRADKAHIPRENMSLPSNMQLNDLSTLRKA-RQTALQIRVGEK--DEEEEEDLSLA--
                 :       ..  :**::*****::*****  .  :.        . :      *  .*

                                        PDZ-binding motif
Human(Hs)    339 -LTSQGFIRLPVLGYIYRVSSVSSDEIWL 366
Mouse(Mm)    338 -LTSRGFIRLPMLGYIYRVSSVSSDEIWL 365
Dog(Cf)      338 -LTSRGFIRLPMLGYIYRVSSVSSDEIWL 365
Frog(Xt)     280 NLDDTGFDRFPLVGYIYKASSVSSDEIWL 308
Fugu(Tr)     355 -YAARGFARYPMVGYIYKVNSTSSEEIWL 382
Medaka(Ol)   355 -YAARGFARYPMVGYIYKVNSTSSEEIWL 382
                 ** * *:::****:..*.**:****
```

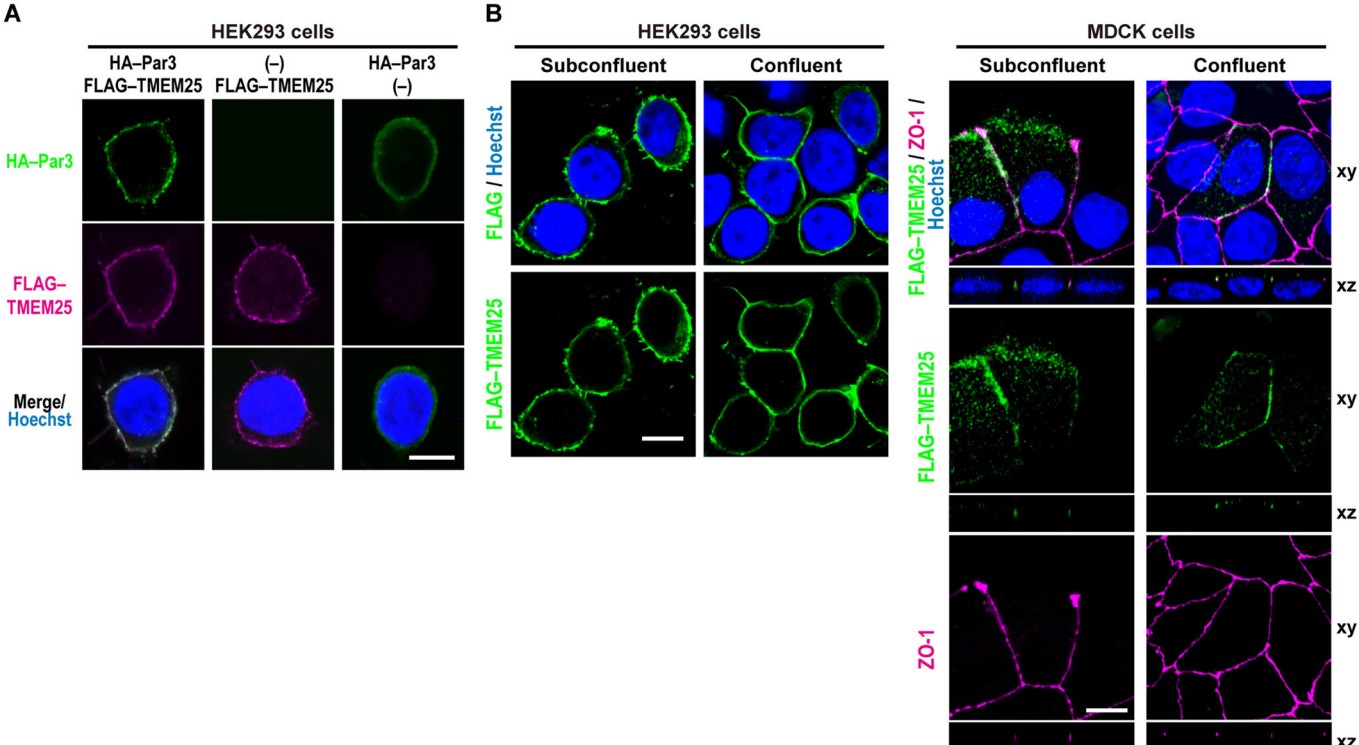

**Figure EV2. Plasma membrane localization of TMEM25 expressed in HEK293 cells and MDCK cells.**

(A) Par3 and TMEM25 expressed in HEK293 cells co-localized at the plasma membrane. HEK293 cells expressing HA–Par3 and/or FLAG–TMEM25 were fixed and stained as indicated. Scale bar, 10 μm. (B) Plasma membrane localization of TMEM25 in confluent and sub-confluent cell cultures. HEK293 cells (left) or MDCK cells (right) expressing FLAG–TMEM25 were grown for 48 h in confluent or sub-confluent cultures, and then fixed and stained with the indicated antibodies and Hoechst. Shown are confocal images (left) or stacked images (xy images in right) of the cells. Cross-sectional z-stack analysis is also shown (xz images in right). Scale bar, 10 μm. Source data are available online for this figure

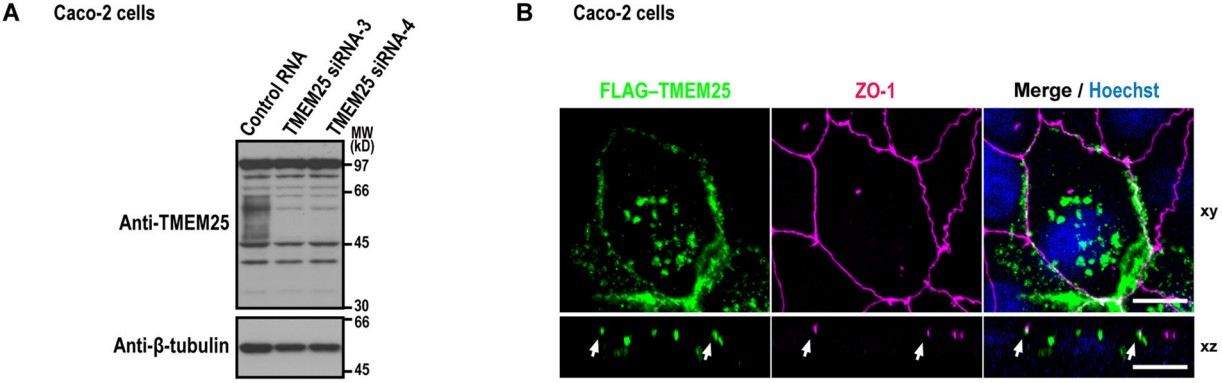

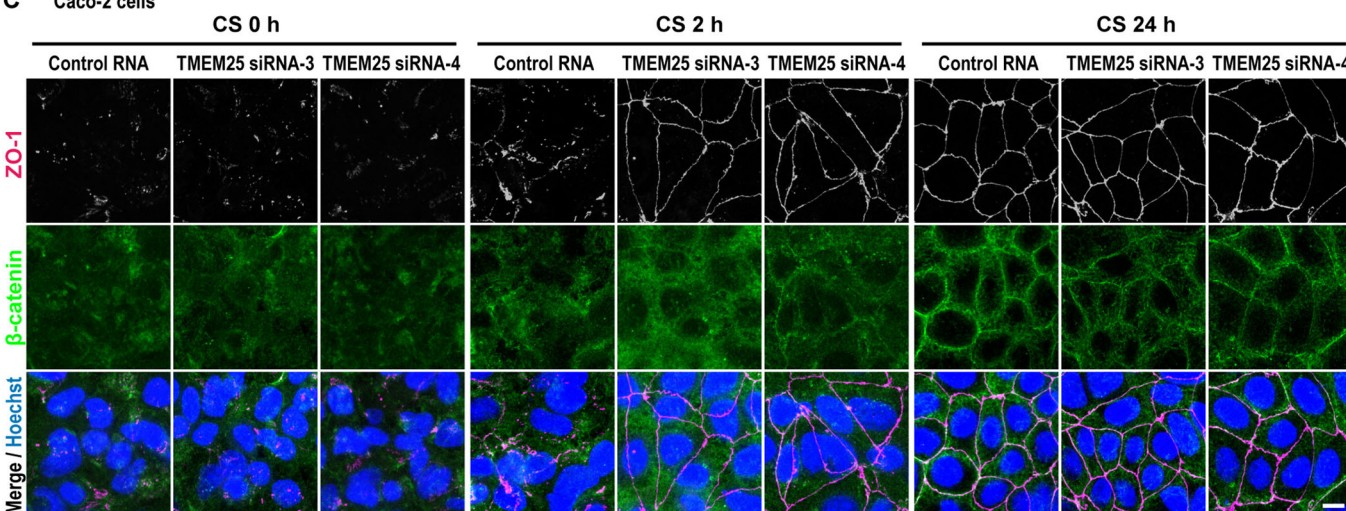

**Figure EV3. Effect of TMEM25 depletion on TJ development in Caco-2 cells.**

(**A**) The expression of endogenous TMEM25 in Caco-2 cells. Proteins in the lysate of Caco-2 cells transfected with control RNA or TMEM25 siRNA were analyzed by immunoblot with the indicated antibodies; a commercially obtained TMEM25 antibody (from Boster) recognizes endogenous TMEM25, which is effectively knocked down by the TMEM25-specific siRNAs. Positions for marker proteins are indicated in kilodaltons (kD). (**B**) Localization of TMEM25 to the TJ in Caco-2 cells. Caco-2 cells transfected with the Tet-on system plasmid pTetOne-FLAG–TMEM25 were grown on the Transwell chamber for 5 days, and treated with 0.2 µg/ml doxycycline for 48 h. Confocal images of the cells were stacked along the z-axis (xy). Cross-sectional z-stack analysis is also shown (xz). Arrows indicate the positions of TJs. Scale bar, 10 µm. (**C**) TJ development in TMEM25-depleted Caco-2 cells after $Ca^{2+}$ switch. Shown are representative images of TMEM25-depleted Caco-2 cells after $Ca^{2+}$ switch. Cells were fixed 0 h, 2 h, or 24 h after $Ca^{2+}$ switch (CS) and stained with the indicated antibodies and Hoechst. Confocal images of the cells were stacked along the z-axis. Scale bar, 10 µm.

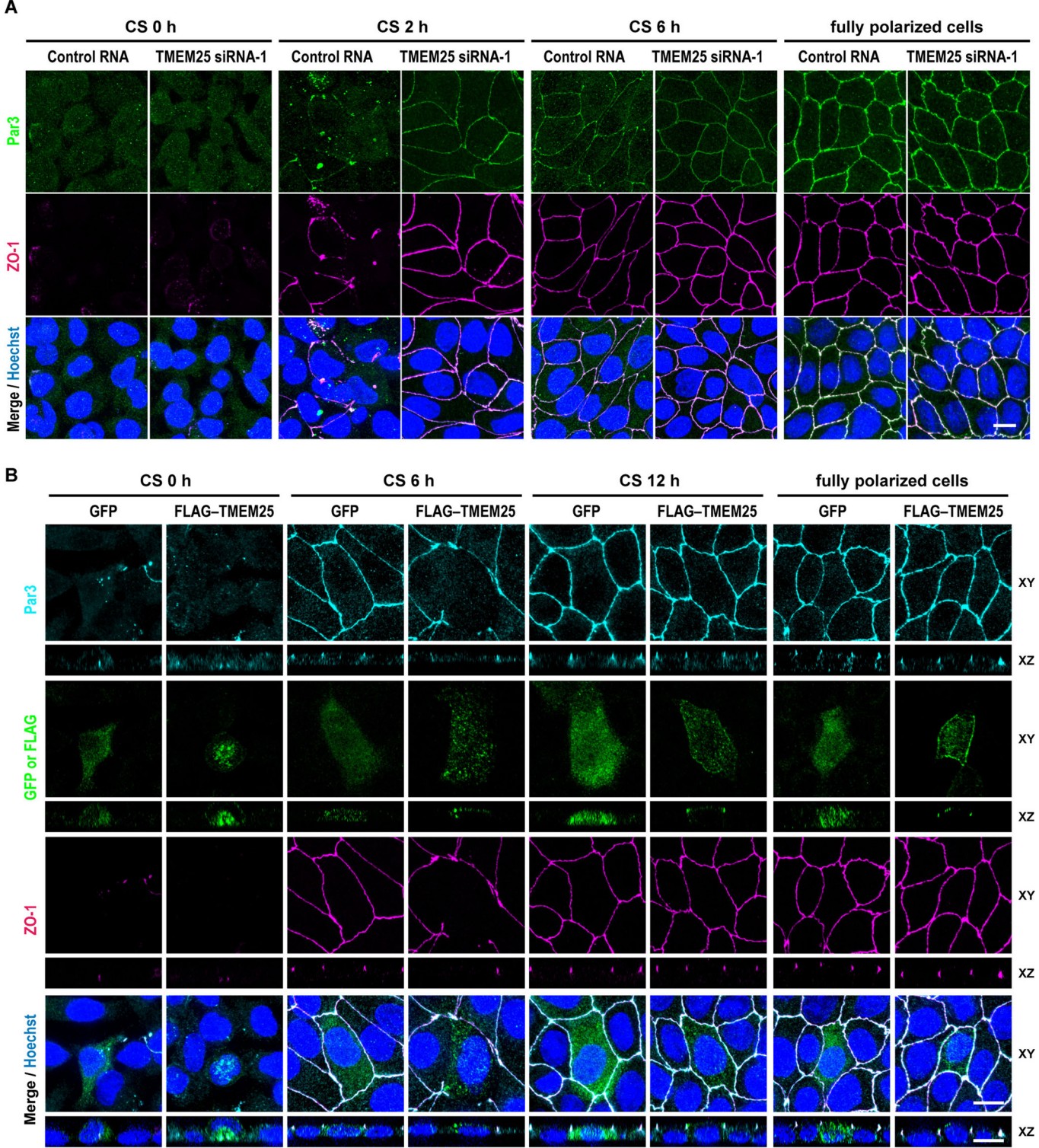

**Figure EV4.  Localization of Par3 in TMEM25-depleted or TMEM25-expressing MDCK cells after Ca$^{2+}$ switch.**

Shown are representative images of TMEM25-depleted MDCK cells (**A**) or the cells expressing GFP alone or FLAG-tagged TMEM25 induced by the Tet-on system in the presence of doxycycline (5 μg/ml) (**B**). Cells were fixed after culture for 72 h (fully polarized cells) or at the indicated time points after Ca$^{2+}$ switch (CS), and stained with the indicated antibodies and Hoechst. Confocal images of the cells were stacked along the z-axis (**A** and xy images in **B**). Cross-sectional z-stack analysis is also shown (xz images in **B**). Scale bar, 10 μm.

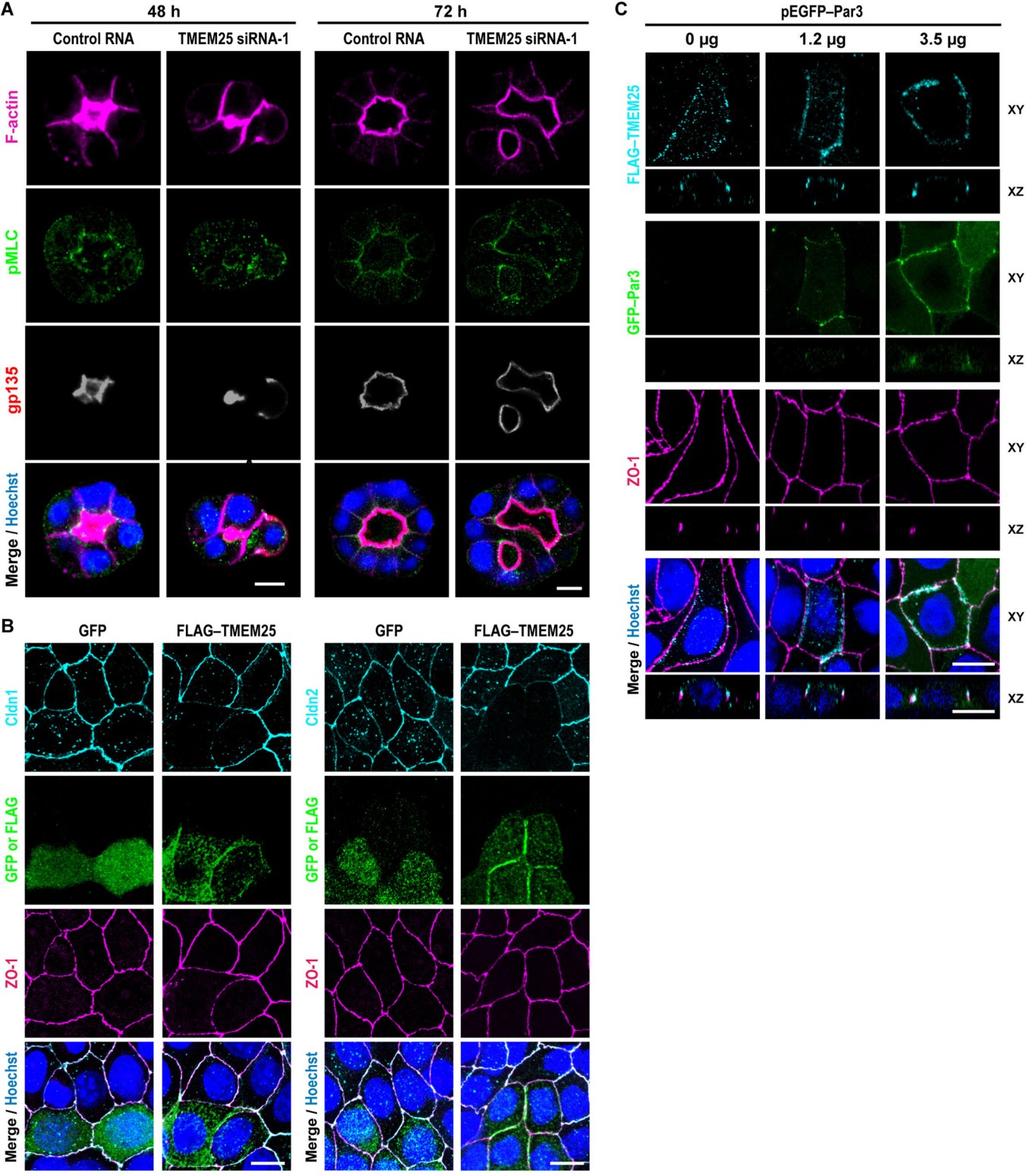

**Figure EV5.   Effect of TMEM25 depletion on the actomyosin organization in MDCK cysts.**

(A) The actomyosin organization in TMEM25-depleted cysts. MDCK cells were transfected with the control RNA or TMEM25 siRNA and grown for 48 h or 72 h in 3D culture and stained with the indicated antibodies, phalloidin (F-actin), and Hoechst. Shown are representative confocal images of cysts. pMLC2, phosphorylated myosin light chain 2. Scale bar, 10 μm. (B) TMEM25-mediated suppression of claudin enrichment at the sites of cell–cell contact in MDCK cells. GFP alone or FLAG–TMEM25 was induced by the Tet-on system in the presence of doxycycline (5 μg/ml) in MDCK cells. Cells were then cultured for 48 h, fixed, and stained with the indicated antibodies and Hoechst. Confocal images of the cells were stacked along the z-axis. Scale bar, 10 μm. (C) Effect of Par3 overexpression on the localization of TMEM25. MDCK cells (2 × 10⁶ cells) were co-transfected with pEGFP–Par3 (0, 1.2, or 3.5 μg) and the Tet-on system plasmid pTetOne-FLAG–TMEM25 (1.5 μg). After culture for 24 h, FLAG–TMEM25 were induced in the presence of doxycycline (1 μg/ml), and cells were cultured for further 48 h, fixed and stained with the indicated antibodies and Hoechst. The amount of pEGFP–Par3 plasmid used for the transfection is shown. Confocal images of the cells were stacked along the z-axis (xy). Cross-sectional z-stack analysis is also shown (xz). Scale bar, 10 μm.

