## [Peer Review File · EMBO Reports]

TMEM25 is a Par3-binding protein that attenuates claudin assembly during tight junction development

Sachiko Kamakura, Junya Hayase, Akira Kohda, Yuko Iwakiri, Kanako Chishiki, Tomoko Izaki, and Hideki Sumimoto
DOI: 10.15252/embr.202356802

Corresponding author(s): Hideki Sumimoto (sumimoto.hideki.851@m.kyushu-u.ac.jp)

Review Timeline:

Submission Date:	10th Jan 23
Editorial Decision:	9th Feb 23
Revision Received:	30th Sep 23
Editorial Decision:	17th Nov 23
Revision Received:	21st Nov 23
Accepted:	22nd Nov 23

Editor: Ioannis Papaioannou / Deniz Senyilmaz Tiebe

Transaction Report:

Dear Prof. Sumimoto,

Thank you for submitting your manuscript for consideration by EMBO reports. It has now been seen by three experts in the field, and we have received the complete set of their reports, which are appended below.

As you will see, the referees acknowledge that the findings are novel, important and potentially interesting. However, they also identify some limitations in the study and the manuscript. Referees #1 and #2 point out that some key experiments should be repeated in more relevant, polarized epithelial cells. In addition, referee #1 raises concerns regarding the experiments suggesting that interaction of the extracellular domains of TMEM25 occurs in cis, as well as about the co-immunoprecipitation experiments with TMEM25 mutants. Referee #3 mentions that the claim that TMEM25 attenuates claudin oligomerization or polymerization should be further confirmed, and that the role of Par3 remains unclear. Furthermore, all referees provide a number of additional suggestions for the improvement of the study and the manuscript.

Given these constructive comments, we would like to invite you to revise your manuscript with the understanding that the referee concerns (as detailed in their reports) must be fully addressed and their suggestions taken on board. Please address all referee concerns in a complete point-by-point response. Acceptance of the manuscript will depend on a positive outcome of a second round of review. It is EMBO reports policy to allow a single round of revision only and acceptance or rejection of the manuscript will therefore depend on the completeness of your responses included in the next, final version of the manuscript. If you have any questions or comments, we can also discuss the revisions in a video chat, if you like.

We realize that it is difficult to revise to a specific deadline. In the interest of protecting the conceptual advance provided by the work, we usually recommend a revision within 3 months (May 8th). Please discuss with me the revision progress ahead of this time if you require more time to complete the revisions.

IMPORTANT NOTE:

We perform an initial quality control of all revised manuscripts before re-review. Your manuscript will FAIL this control and the handling will be DELAYED if the following APPLIES:

- 1) If a data availability section providing access to data deposited in public databases is missing.
- 2) If your manuscript contains statistics and error bars based on $n=2$. Please use scatter plots in these cases. No statistics should be calculated if $n=2$.

- 1) a .docx formatted version of the manuscript text (including legends for main figures, EV figures and tables). Please make sure that the changes are highlighted to be clearly visible.
- 2) individual production quality figure files as .eps, .tif, .jpg (one file per figure). Please download our Figure Preparation Guidelines (figure preparation pdf) from our Author Guidelines pages <https://www.embopress.org/page/journal/14693178/authorguide> for more info on how to prepare your figures.
- 3) a .docx formatted letter INCLUDING the reviewers' reports and your detailed point-by-point responses to their comments. As part of the EMBO Press transparent editorial process, the point-by-point response is part of the Review Process File (RPF), which will be published alongside your paper unless you opt out of this (please see below for further information).
- 4) a complete author checklist, which you can download from our author guidelines (<<https://www.embopress.org/page/journal/14693178/authorguide>>). Please insert information in the checklist that is also reflected in the manuscript. The completed author checklist will also be part of the RPF.
- 5) Please note that all corresponding authors are required to supply an ORCID ID for their name upon submission of a revised manuscript (<<https://orcid.org/>>). Please find instructions on how to link your ORCID ID to your account in our manuscript tracking system in our Author guidelines (<<https://www.embopress.org/page/journal/14693178/authorguide#authorshipguidelines>>)

6) We replaced Supplementary Information with Expanded View (EV) Figures and Tables that are collapsible/expandable online. A maximum of 5 EV Figures can be typeset. EV Figures should be cited as 'Figure EV1, Figure EV2' etc... in the text and their respective legends should be included in the main text after the legends of regular figures.

7) Before submitting your revision, primary datasets produced in this study need to be deposited in an appropriate public database (see < <https://www.embopress.org/page/journal/14693178/authorguide#dataavailability>>).

Specifically, we would kindly ask you to provide public access to the following datasets/data:

- Liquid chromatography-tandem mass spectrometry data

The accession numbers and database should be listed in a formal "Data Availability " section (placed after Materials and Methods) that follows the model below (see also < <https://www.embopress.org/page/journal/14693178/authorguide#dataavailability>>):

Data availability

*** Note: all links should resolve to a page where the data can be accessed. ***

*** Note: the Data Availability Section is restricted to new primary data that are part of this study. ***

8) We now request authors to consider both actual and perceived competing interests. Please review the new policy (<<https://www.embopress.org/competing-interests>>) and update your competing interests statement if necessary. Please name this section 'Disclosure and competing interests statement' and place it after the Acknowledgements section.

9) Figure legends and data quantification:

- the name of the statistical test used to generate error bars and P values,
- the number (n) of independent experiments (please specify technical or biological replicates) underlying each data point,
- the nature of the bars and error bars (s.d., s.e.m.)
- If the data are obtained from n {less than or equal to} 2, use scatter plots showing the individual data points.

10) We now request the publication of original source data with the aim of making primary data more accessible and transparent to the reader. Our source data coordinator will contact you to discuss which figure panels we would need source data for and will also provide you with helpful tips on how to upload and organize the files.

11) Our journal encourages inclusion of *data citations in the reference list* to directly cite datasets that were re-used and obtained from public databases. Data citations in the article text are distinct from normal bibliographical citations and should directly link to the database records from which the data can be accessed. In the main text, data citations are formatted as follows: "Data ref: Smith et al, 2001" or "Data ref: NCBI Sequence Read Archive PRJNA342805, 2017". In the Reference list, data citations must be labeled with "[DATASET]". A data reference must provide the database name, accession

number/identifiers and a resolvable link to the landing page from which the data can be accessed at the end of the reference. Further instructions are available at <<https://www.embopress.org/page/journal/14693178/authorguide#referencesformat>>.

12) Please also note our reference format:

<<http://www.embopress.org/page/journal/14693178/authorguide#referencesformat>>.

13) We now use CRediT to specify the contributions of each author in the journal submission system. CRediT replaces the author contribution section, which should be removed from the manuscript. Please use the free text box to provide more detailed descriptions. See also guide to authors:

<<https://www.embopress.org/page/journal/14693178/authorguide#authorshipguidelines>>.

14) As part of the EMBO publications' Transparent Editorial Process, EMBO reports publishes online a Review Process File to accompany accepted manuscripts. This File will be published in conjunction with your paper and will include the referee reports, your point-by-point response and all pertinent correspondence relating to the manuscript.

You can opt out of this by letting the editorial office know (emboreports@embo.org). If you do opt out, the Review Process File link will point to the following statement: "No Review Process File is available with this article, as the authors have chosen not to make the review process public in this case."

I look forward to seeing a revised version of your manuscript when it is ready. Please let me know if you have any questions or comments regarding the revision.

Yours sincerely,

Ioannis Papaioannou, PhD
Editor
EMBO reports

Referee #1:

The manuscript by Sumimoto and colleagues describes the identification of the immunoglobulin superfamily member TMEM25 as a Par3-interacting protein and novel component of tight junctions (TJs). The authors characterize the biochemical characteristics of TMEM25 to some extent. In functional assays, the authors identify a negative regulatory role for TMEM25 in TJ formation as depletion of TMEM25 accelerates barrier formation after Ca²⁺ switch whereas ectopically expressed TMEM25 seems to impair junction formation. Also, depletion of TMEM25 inverts apico-basal polarity. Besides Par3, claudin-1 and claudin-2 are additional binding partners for TMEM25. Interestingly, TMEM25 seems to prevent Claudin oligomerization providing a potential mechanism through which TMEM25 negatively regulates the barrier function. Finally, ectopically expressed Par3 impairs/inhibits the interaction between TMEM25 and Cldn1, and the findings suggest that Par3 binds to TMEM25 to prevent TMEM25 - claudin interactions to facilitate barrier formation. This is a very interesting study with the potential to provide new insights into the mechanisms underlying claudin-mediated tight junction regulation. All experiments are well performed and nicely controlled.

As a more principal notion, the authors propose a model on the regulation of claudin homo-/hetero multimerization by TMEM25 which is based on biochemical experiments performed in HEK293 cells. Two issues remain: First, although the biochemical experiments are mostly convincing, it is unclear how representative the data are, as quantification of the Western blot data is missing throughout the manuscript. More importantly, all biochemical experiments were performed in non-polarizing HEK293 cells, which do not have tight junctions. It, thus, remains questionable, if the observations on the molecular mechanisms are applicable to polarized epithelial cells. A few key experiments should be performed in polarized epithelial cells. MDCKII cells should provide a suitable model system.

Specific comments

1. The authors describe a direct and PDZ domain-mediated interaction between TMEM25 and Par3 (Fig. 1C, D). While the interaction data as such are convincing, they do not allow for the conclusion that the TMEM25 interacts with Par3 at the TJs of

- polarized epithelial cells. The authors should demonstrate the interaction with endogenous proteins rather than with ectopically expressed proteins and using a polarizing epithelial cell line instead of poorly polarizing HEK293 cells.
- In Fig. 1G, the authors show a TJ localization of Flag-TMEM25 with endogenous ZO-1 and Par3, which is convincing. Again, the localization of endogenous TMEM25 at TJs should be demonstrated. The authors use ectopically expressed TMEM25 arguing that no antibodies are available against endogenous TMEM25 in MDCK cells. However, kidney- and intestine-derived human epithelial cell lines which nicely polarize do exist (e.g. Caco2, T84, Eph4, HK-2). Antibodies against human TMEM25 are available.
 - The experiments shown in Fig. 2A and B suggest a direct interaction of the extracellular domains of TMEM25. The authors conclude that the interaction occurs in cis. Why? The interaction could as well be in trans. The soluble MBP-TMEM25-ECR fusion protein would most likely allow for both cis and trans interaction with immobilized GST-TMEM25-ECR. The authors also mention that the CoIPs shown in Fig. 2B were performed with cells cultured at low density. Can they exclude that the cells have contacts? A proximity ligation assay (PLA) performed on cells in isolation could tell if the cis interaction exists. Alternatively, CoIP experiments with single cell suspension cultures could be performed. Based on the absence of a CoIP interaction when cells are grown at confluency (Fig. 2E), the authors also conclude that a trans interaction does not exist. However, why should the cis-interaction be absent when the cells have reached confluency? Wouldn't the loss of an interaction at confluency rather suggest that TMEM25 is internalized? The authors should perform immunofluorescence studies to visualize TMEM25 at subconfluent and confluent growth conditions. Also, such experiments should be performed in polarized MDCK cells to address the significance for tight junction physiology.
 - The CoIP experiments with the TMEM25- Δ ECR and TMEM25- Δ ICR mutants (Fig. 2C) are confusing. Deletion of the entire extracellular region (TMEM25- Δ ECR) does only slightly impair the homophilic interaction. Since deletion of the ICR has no effect, the authors conclude that the transmembrane segment is involved. This is, however, highly speculative. To prove the involvement of the transmembrane segment, CoIPs would have to be performed with swapping mutants using the transmembrane segment of a protein that does not form cis dimers.
 - The immunofluorescence signals at cell junctions of TMEM25 siRNA-treated cells (Fig. 3) should be quantified.
 - The authors find that strong overexpression of TMEM25 (but not TMEM25/ Δ C) in MDCKII cells impairs the localization of ZO-1 at cell junctions (Fig. 4A). However, it is not clear if the IF images present single confocal sections or projected views. It thus, cannot be judged, if the loss of ZO-1 in TMEM25-expressing cells is due to mislocalization of ZO-1 to the basolateral membrane. The authors should show confocal XZ-sections of TMEM25-transfected cells to visualize the localization of ZO-1 along the lateral cell border. Also, it is not clear why permeability assays (TER measurements) were not performed (analogous to Fig. 3A). If the number of cells expressing TMEM25 is too small, stable cell lines can be generated and eventually subcloned which will allow the analysis of TMEM25 overexpression not only at the single cell level but also at the population level.
 - Similar to the conclusion of a homophilic TMEM25 interaction in cis, the authors conclude from CoIP experiments of HEK293 cells grown at low density that TMEM25 interacts with claudin-1 in cis (Fig. 5). Again, the notion "low density" is poorly defined. Can the authors exclude the possibility that cells have contacts to other cells? Since a lateral association of TMEM25 with claudins is a very important observation, the lateral (cis) association should be more vigorously demonstrated. The authors should perform proximity ligation assays on single isolated cells in the absence of cell-cell contacts.
 - The authors propose that TMEM25 prevents claudin cis- and trans-oligomerization (Fig. 6). Like the previous experiments, these experiments were performed in non-polarizing HEK293 cells which do not develop tight junctions. Is the same effect observed in polarizing MDCKII cells? The authors could express HA-TMEM25 together with Flag-Cldn1 in MDCKII cells and analyze claudin hetero-oligomerization of ectopic Flag-Cldn1 and endogenous Cldn2. In a complementary approach, the authors could downregulate TMEM25 in MDCKII cells and analyze the hetero-oligomerization between Flag-Cldn1 and endogenous Cldn2 (or a differently tagged Cldn2).

Additional points

- The terms "membrane-integrated constituents" and "soluble regulatory proteins" (first sentence of the abstract) is confusing. The more common terminology is "integral membrane proteins" and "peripheral membrane proteins" / "cytoplasmic scaffolding proteins", respectively.
- The plasmid constructs should be described in more detail. For example, it is not clear which parts of TMEM25 have been cloned into pGEX and pMAL plasmids for expression in E.coli. The authors only mention "The cDNA fragments of various regions of these proteins were amplified by PCR...". For recapitulation of the findings, the regions of the protein fragments must be indicated. Meanwhile, this is standard for most journals.
- The term "intracellular interaction" is misleading as it suggests an interaction in the cytoplasm. The term "cis-interaction" and "trans-interaction", which are commonly used to describe protein interactions in the same membrane and on apposing membranes, respectively, should be used instead.
- In the legend to Fig. 4B, the authors claim to have analyzed "n > 75 cells/experiment". It probably must mean "cysts/experiment". Also, in the statistical evaluation the authors should distinguish between multiluminal cysts and cells with inverted apico-basal polarity. Inverted polarity and multiluminal cysts are phenotypes of very different molecular mechanisms and thus should be separately evaluated.
- In Fig. 6A, TMEM25 is very strongly expressed in the samples co-transfected with Cldn2 (Lysate sample). However, the CoIP sample (IP: Flag; Blot: HA) shows no stronger signal than for Cldn1. Is the strong signal observed in the lysates representative?
- Typos: TetOn instead of TetOne (p.18)

Referee #2:

TMEM25 is a Par3-binding protein that attenuates claudin assembly during tight junction development.
Sachiko Kamakura, Junya Hayase, Akira Kohda, Yuko Iwakiri, Kanako Chishiki, Tomoko Izaki, & Hideki Sumimoto

This study from Kamakura et al. proposes a mechanism by which TMEM25 negatively regulates claudin assembly via Par3 during tight junction formation. Using imaging and biochemical assays, authors showed that TMEM25 is a tight junction protein that negatively regulates claudin assembly and thereby tight junction formation. Additionally, Par3 binding to TMEM25 prevented TMEM25-Claudin interaction and enhanced tight junction formation. This is an interesting study with novel findings; however, some concerns are listed below:

Major concerns:

1. Most of the work shown here are performed in MDCK cells. Given that, TMEM25 is highly expressed in both kidney and intestinal epithelial cells, authors should test key findings in an intestinal epithelial cell line such as CaCo2 or T-84.
2. Based on biochemical data, authors report that TMEM25 overexpression prevents claudin oligomerization. What's the effect of TMEM25 OE on structure of TJ strands in MDCK cells, or in HEK293 cells co-expressing fluorescently tagged claudin-1 and TMEM25?
3. In figure 3, authors show that TMEM25 negatively regulates TJ formation. What's the effect of TMEM25 knockdown on the lateral localization of TJ and AJ proteins in MDCK at baseline? Is the polarity of MDCK cells maintained on TMEM25 knockdown?
4. In figure 4, authors report that loss of TMEM25 results in abnormal cysts formation. Is the abnormal cysts formation due to defective actomyosin organization in the cells? Authors should co-stain for F-actin and pMLC in control and TMEM25 KD cells.
5. What's the effect of TMEM25 OE or KD on Par3 localization following calcium switch and in mature epithelial cells?

Minor concerns:

1. What is the effect of TMEM25 knockdown on macromolecular flux following a calcium switch?

Referee #3:

In this article, Sachiko Kamakura et. al found TMEM25 as PAR3 binding protein that regulates tight junction formation by attenuating the claudin assembly. The work contains solid biochemical experiments showing Cis- but no Trans- interaction between TMEM25, Cis- and Trans-interactions with Claudin-1 and 2, and also the dependence on the ECR and C terminal binding to Par3. Functional studies in MDCK tissue suggest that TMEM25 inhibits TJ-formation by blocking Claudin polymerization.

This is very nice work and the findings are important for our understand how TJ formation is controlled. I have several comments that should be addressed before publication:

Main points:

The authors propose that TMEM25 attenuates claudin oligomerization or polymerization. This should be confirmed by claudin1 and claudin2 staining in MDCK cells and in the HEK overexpression system. In particular, in the HEK system claudin strand formation can be nicely seen by confocal microscopy. Does co-expression of TMEM25 alter claudin strand formation?

The role of Par3 remains rather unclear. The authors suggest a concentration dependent function. Could the authors check whether the localization of TMEM25 is altered by increasing Par3 concentrations? One explanation could be that at high Par3 concentration TMEM25 is does not accumulate at the TJ, because it interacts with PAR3 at other regions (competition).

Minor points:

In figure1D, why is the higher band more likely to be pulled down by HA-Par3? Is glycosylation important for Par3 binding? There is a typo in the introduction, line 4: adherence junction should be adherens junction.

Manuscript Number: EMBOR-2023-56802V1

Title: TMEM25 is a Par3-binding protein that attenuates claudin assembly during tight junction development

Dear Dr. Ioannis Papaioannou, Editor of EMBO Reports:

Thank you very much for your kind e-mail of February 9th, 2023, regarding the decision of our manuscript submitted to *EMBO Reports* (EMBOR-2023-56802V1).

We are also very grateful for your kind consideration about the extension of the deadline.

We thank you and the referees for an excellent and thorough review of our manuscript.

We are very happy to hear that you would consider a revised version of the manuscript, if we are able to fully address the referee concerns and take their suggestions on board.

According to the suggestions raised by referees, we have performed many experiments and revised the manuscript.

A list of changes made and responses to each of points raised by referees are included below.

We appreciate both your help and those of the referees' to improve this paper.

We hope that the paper is now suitable for publication in *EMBO Reports*.

Yours faithfully,

Hideki Sumimoto, M.D., Ph.D.

Professor

Department of Biochemistry,

Kyushu University Graduate School of Medical Sciences

3-1-1 Maidashi, Higashi-ku, Fukuoka 812-8582, Japan

Tel: 81(Japan)-92-642-6096

E-mail: sumimoto.hideki.851@m.kyushu-u.ac.jp

Answers to Referee 1

We thank Referee 1 very much for an excellent and thorough review of our manuscript. According to the suggestions raised by Referee 1, we have performed many experiments and revised the manuscript.

Responses to each of points raised by Referee 1 are included below.

We appreciate the referee's help to improve this paper.

We hope that the paper is now suitable for publication in *EMBO Reports*.

Principle notes:

(1) Referee 1 suggested that the authors should quantify the Western blot data.

According to the suggestion, we have quantified the Western blot data, especially important ones, which have been presented in Figure 2E, Figure 6A, Figure 6B, and Figure 6F of the revised manuscript.

(2) Referee 1 suggested that although all biochemical experiments were performed in non-polarized HEK293 cells, a few key experiments should be performed in polarized epithelial cells such as MDCK cells.

We completely agree with the notion. Indeed, we have performed many experiments and the details of the results obtained are mentioned below.

Specific comments:

(1) Referee 1 suggested that the authors should demonstrate the interaction between endogenous Par3 and endogenous TMEM25 using a polarized epithelial cell line instead of poorly polarizing HEK 293 cells.

We completely agree with the importance of the demonstration for the interaction with endogenous proteins in a polarized epithelial cell line. For this purpose, an excellent antibody for TMEM25 is required. Actually, we prepared three monoclonal antibodies raised against the extracellular region (residues 27–220) of TMEM25 and two polyclonal antibodies raised against the extracellular region (residues 27–220) or the C-terminal peptide (residues 295–308) of TMEM25. Unfortunately, all the antibodies do not react with endogenous TMEM25 in immunoblot analysis, immunoprecipitation, or immunostaining of various epithelial cells (MDCK cells, Coco-2 cells, and Eph4 cells), although they specifically recognize overexpressed TMEM25, indicative of a low affinity. According to your suggestion, we purchased four commercially-available anti-TMEM25 antibodies and tested them. Again, they react well with overexpressed TMEM25 but fail to recognize endogenous TMEM25 in immunoprecipitation and immunostaining of MDCK cells and Coco-2 cells. Only one antibody, which was obtained from Boster, but not other three antibodies can be used in immunoblot analysis of MDCK cells and Caco-

2 cells, albeit with many non-specific bands (please see Figure 1F and Figure EV3A). Collectively, no anti-TMEM25 antibody is available for immunoprecipitation or immunostaining at present.

Another way to detect the interaction between the endogenous proteins was to use an anti-Par3 antibody for immunoprecipitation. We have performed many experiments using several anti-Par3 antibodies available but found that there are no anti-Par3 antibody suitable for immunoprecipitation.

Hence, we have expressed FLAG-tagged TMEM25 in MDCK cells and shown that endogenous Par3 was co-precipitated with FLAG-TMEM25. The result has been presented in Figure 1H and described in the Results section of the revised manuscript (page 6, lines 5–8 from the bottom).

(2) Referee 1 suggested that the localization of endogenous TMEM25 at TJs should be demonstrated.

Unfortunately, as mentioned above in detail, no anti-TMEM25 antibody is presently unavailable for immunostaining of endogenous TMEM25, to our knowledge. Actually, for example, the TJs in Caco-2 cells were weakly stained with two of the commercially available antibodies; however, in both cases, the signal was not decreased by siRNA-mediated depletion of TMEM25.

(3) Referee 1 suggested that the authors should perform a proximity ligation assay (PLA) to show that homophilic interaction of TMEM25 occurs in *cis*.

We thought that it is a very nice suggestion. Thus, according to the suggestion, we have performed the PLA experiments and found that the PLA signal was detected on the cell surface solely in a HEK293 cell expressing both HA-TMEM25 and FLAG-TMEM25 but not in a cell with either protein, indicating that TMEM25 self-associates in a *cis* manner at the plasma membrane. This result has been presented in Figure 2C and described in the Results section of the revised manuscript (page 7, lines 8–15).

As shown in the present Figure 2G (corresponding to Figure 2E of the original manuscript), when HEK293 cells expressing solely FLAG-TMEM25 and HEK293 cells expressing solely HA-TMEM25 were co-cultured to confluence, the interaction between FLAG-TMEM25 and HA-TMEM25 was not detected by CoIP experiments. On the other hand, when HEK293 cells co-expressing FLAG-TMEM25 and HA-TMEM25 were used, the interaction was detected even at low cell density (Figure 2B).

To exclude the possibility that TMEM25 is internalized at confluency, Referee 1 suggested that the author should perform immunofluorescence studies to visualize

TMEM25 at subconfluent and confluent growth conditions, and also suggested that similar experiments should be performed in polarized MDCK cells.

According to the suggestion, we have performed experiments and found that TMEM25 was correctly presented on the surface of these cells within confluent monolayers. This result has been presented in Figure EV2B and described in the Results section of the revised manuscript (page 8, lines 3–6).

(4) To prove the involvement of the transmembrane segment in the homophilic interaction of TMEM25, Referee 1 suggested that the authors should perform CoIP experiments with swapping mutants using the transmembrane segment of a protein that does not form cis dimer.

We completely agreed with the suggestion, and thus we have constructed several mutant proteins. Among them, we have selected a mutant TMEM25 with replacement of the transmembrane segment with the corresponding region of JAM-A, a protein incapable of binding to TMEM25 (as shown in Figure 2B), and tested its ability to interact with TMEM25-FL (wild-type, full-length TMEM25). The reason for the selection of JAM-A is as follows: the JAM-A chimera protein is well expressed in cells compared with other mutant proteins; JAM-A is a tight junction (TJ) protein that belongs to the Ig superfamily like TMEM25 but is unable to interact with TMEM25, and thus seems to be suitable as a control; and, although JAM-A is able to form a homodimer, it is known that the extracellular region but not the transmembrane region is involved in JAM-A homodimer formation.

The mutant protein (TM-JAM chimera) interacted with wild-type TMEM25 (TMEM25-FL) more weakly compared with self-interaction of wild-type proteins, indicating that the transmembrane segment is also involved in self-association of TMEM25. The result has been presented in Figure 2E and described in the Results section of the revised manuscript (page 7, lines 16–20).

(5) Referee 1 suggested that the immunofluorescence signals at cell junctions of TMEM25 siRNA-treated cells should be quantified (Figure 3).

According to the suggestion, we have quantified the signal using line plot analysis. The results have been presented at the bottom of Figures 3C, 3D, and 3E of the revised manuscript.

(6) Referee 1 suggested that the authors should clarify whether the IF images of Figure 4A present single confocal sections or projected views, and should show that confocal XZ-sections of TMEM25-transfected cells to visualize the localization of ZO-1 along the lateral cell border.

According to the suggestion, we have mentioned that the IF images present projected views in the legend to Figure 4A of the revised manuscript. We have shown that confocal XZ-sections of TMEM25-transfected cells in Figure 4A of the revised manuscript.

The referee also suggested that, to further know the effect of overexpression of TMEM25 in MDCK cells, the authors should perform permeability assays (TER measurements).

As mentioned by the referee, the number of cells expressing TMEM25 is too small when usual lipofection methods are used for MDCK cells. Hence, we have prepared Lentivirus vectors for efficient expression of TMEM25, which enabled us to analyze the effect of TMEM25 overexpression at the population level. As expected, overexpression of full-length TMEM25 delayed calcium-induced TER increase. This result has been presented in Figure 4B and described in the Results section of the revised manuscript (page 9, lines 7–11).

(7) Referee 1 mentioned that since a lateral (*cis*) association of TMEM25 with claudins is a very important observation, the lateral (*cis*) association should be more vigorously demonstrated, and suggested that the authors should perform PLAs on single isolated cells in the absence of cell–cell contacts.

We completely agreed with the suggestion, we have performed the PLA and found that the PLA signal was detected on the cell surface solely in a cell expressing both TMEM25–HA and FLAG–claudin-1, indicating that TMEM25 interacts with claudin-1 in a *cis*-configuration at the plasma membrane. This result has been presented in Figure 5D and described in the Results section of the revised manuscript (page 10, lines 18–22).

(8) As indicated by Referee 1, we propose that TMEM25 prevents claudin *cis*- and *trans*-oligomerization on the basis of ColP experiments using non-polarized HEK293 cells which do not develop tight junctions (Figure 6). Thus, Referee 1 asked whether the same effect is observed in polarizing MDCK cells.

We thought that this is a very important question, and thus we have performed many ColP experiments using various types of detergents but failed to solubilize oligomerized claudins.

Consistent with the failure, many investigators have reported that the ColP experiments for claudins do not work in epithelial cells such as MDCK cells and Caco-2 cells, because claudins are polymerized into strands which are highly insoluble in detergents (for example, Itoh *et al.* *J. Cell Biol.* 147, 1351 (1999); Shen *et al.* *J. Cell Sci.* 119, 2095 (2006); Raleigh *et al.* *Mol. Biol. Cell* 21, 1200 (2010)).

On the other hand, it is known that when expressed in claudin-deficient cells such as HEK293 cells, claudins are able to form tight junction-like strands. Using HEK293 cells,

we have found that ectopically-expressed claudin1 or claudin2 was enriched at contact sites between two claudin-expressing HEK293 cells and the enrichment was effectively suppressed by co-expression of TMEM25, indicating that TMEM25 attenuates claudin trans-interaction and subsequent strand formation. The result has been presented in Figure 6E and described in the Results section of the revised manuscript (page 11, lines 11–27).

Furthermore, we have also found that overexpression of TMEM25 in MDCK cells inhibited accumulation of endogenous claudin1 and claudin2 at cell–cell contact regions, which has been presented in Figure EV5B and described in the Results section of the revised manuscript (page 11, lines 24–27).

Additional points:

(1) Referee 1 pointed out that the term “membrane-integrated constituents” and “soluble regulatory proteins” (the first sentence of the abstract) is confusing, and thus suggested that the more common terminology, i.e., “integral membrane proteins” and “peripheral membrane proteins” / “cytoplasmic scaffolding proteins”, respectively, should be used. According to the suggestion, we have modified the first sentence of the abstract (page 2, lines 2–3) as follows: The tight junction (TJ) in epithelial cells is formed by integral membrane proteins and cytoplasmic scaffolding proteins.

(2) Referee 1 suggested that the plasmid constructs should be described in more detail, especially the precise regions of the proteins used.

According to the suggestion, we described the amino acid residue numbers of all of the proteins or protein fragments used in the present study in the plasmids section of the Materials and methods section of the revised manuscript (page 16, lines 2–32).

(3) Referee 1 kindly pointed out that the term “intracellular interaction” is misleading as it suggests an interaction in the cytoplasm, and thus suggested that the term “cis-interaction” and “trans-interaction” should be used instead.

According to the suggestion, we used the term “cis-interaction” and “trans-interaction” instead of “intracellular interaction” in the revised manuscript.

(4) Referee 1 kindly pointed out that “ $n \geq 75$ cells/ experiment” must be “ $n \geq 75$ cysts/ experiment” in the legend to the present Figure 4C (corresponding to Figure 4B in the original manuscript)

According to the suggestion, we have corrected this in the legend to Figure 4C of the revised manuscript.

Referee 1 also suggested that in the statistical evaluation of data of the present Figure 4C (corresponding to Figure 4B in the original manuscript), “cysts with multiple lumens” and “cysts containing cells with inverted polarity” should separately evaluated.

According to the suggestion, we have evaluated them separately in Figure 4C of the revised manuscript.

(5) In Figure 6A of the original manuscript, Referee 1 pointed out that TMEM25 is very strongly expressed in the samples co-transfected with Cldn2, but the CoIP sample (IP: FLAG; Blot: HA) shows no stronger signal than for Cldn1. Thus, the referee asked whether the strong signal observed in the lysates is representative.

First of all, we would like to apologize for our careless mistake about the data in the previous Figure 6A. Since the samples (the lysates) at that time was stored in a deep freezer, we have performed immunoblot analysis using the stored samples several times, and found that the expression level of TMEM25 was not different between Cldn1-coexpressing and Cldn2-coexpressing cells. We have also performed the same experiment using newly transfected cells and confirmed that there is no difference in TMEM25 expression between Cldn1-coexpressing and Cldn2-coexpressing cells. Thus, we have replaced the previous panels of immunoblot with Anti-HA (TMEM25) using the lysate with the new panels that has been obtained by the present analysis in Figure 6A of the revised manuscript.

(6) Referee 1 kindly pointed out a typo: TetOn instead of TetOne (p18).

We have corrected this error in the Materials and methods section of the revised manuscript (page 22, line 3).

Answers to Referee 2

We thank Referee 2 very much for an excellent and thorough review of our manuscript. According to the suggestions raised by Referee 2, we have performed many experiments and revised the manuscript.

Responses to each of points raised by Referee 2 are included below.

We appreciate the referee's help to improve this paper.

We hope that the paper is now suitable for publication in *EMBO Reports*.

Major concerns:

(1) As indicated by Referee 2, most of the work shown here are performed in MDCK cells. Thus, the referee suggested that the authors should test key findings in an intestinal epithelial cell line such as Caco-2 or T-84.

According to the suggestion, we have performed many experiments and obtained the following results.

First, we have detected the TMEM25 protein is expressed in Caco-2 cells by immunoblot analysis, which has been presented in Figure EV3A and described in the Results section of the revised manuscript (page 6, lines 12–15).

Second, we have demonstrated, using the TetOn system, that TMEM25 localizes to the tight junction (TJ) also in Caco-2 cells, which has been presented in Figure EV3B and described in the Results section of the revised manuscript (page 6, lines 18–23).

Third, TMEM25 knockdown in Caco-2 cells leads to an accelerated TJ assembly, which has been presented in Figure EV3C and described in the Results section of the revised manuscript (page 8, lines 2–3 from the bottom).

(2) Referee 2 asked what is the effect of TMEM25 OE on structure of TJ strands in MDCK cells or in HEK293 cells co-expressing fluorescently tagged claudin-1 and TMEM25. The referee suggested that the authors should answer the question whether co-expression of TMEM25 alters claudin strand formation using the HEK293 overexpression system. To answer these questions, we have performed many experiments and found that ectopically-expressed claudin1 or claudin2 was enriched at contact sites between two claudin-expressing HEK293 cells and the enrichment was effectively suppressed by co-expression of TMEM25, indicating that TMEM25 attenuates claudin trans-interaction and subsequent strand formation. The result has been presented in Figure 6E and described in the Results section of the revised manuscript (page 11, lines 11–27).

Furthermore, we have also found that overexpression of TMEM25 in MDCK cells inhibited accumulation of endogenous claudin1 and claudin2 at cell–cell contact regions, which has been presented in Figure EV5B and described in the Results section of the revised manuscript (page 11, lines 24–27).

(3) Referee 2 asked what is the effect of TMEM25 knockdown on the lateral localization of TJ and AJ proteins in MDCK cells at baseline, and also asked whether the polarity of MDCK cells is maintained on TMEM25 knockdown.

To answer the question, we have added xz images in Figure 3C of the revised manuscript. The images demonstrate that 2 h after calcium switch, TMEM25 depletion facilitated the formation of continuous junctions with the correct lateral localization of the TJ protein ZO-1 and the AJ protein E-cadherin. These findings indicate that TMEM25 depletion leads to an accelerated TJ development in cells that maintain apico-basal polarity. This has been presented in Figure 3C and Figure EV4A, and described in the Results section of the revised manuscript (page 8, lines 5–8 from the bottom).

(4) To answer the question whether abnormal cyst formation induced by loss of TMEM25 is due to defective actomyosin organization in the cells, Referee 2 suggested that the authors should co-stain for F-actin and pMLC in control and TMEM25 KD cells.

According to the suggestion, we have co-stained for F-actin and pMLC in control and TMEM25 KD cells that constitute cysts in 3D culture and found that pMLC colocalized well with F-actin even in TMEM25 KD cysts, which has been shown in Figure EV5A of the revised manuscript. The result suggests that actomyosin organization is not strongly disturbed by depletion of TMEM25, which has been described in the Results section of the revised manuscript (page 9, lines 2–7 from the bottom).

(5) Referee 2 asked what is the effect of TMEM25 OE and KD on Par3 localization following calcium switch and in mature epithelial cells.

To address this question, we have performed several experiments and obtained the following results.

First, TMEM25 OE delayed Par3 localization to cell–cell contact sites following calcium switch, which is consistent with a delayed TJ development; and TMEM25 OE did not affect correct localization of Par3 to the TJ in fully polarized cells. These results have been presented in Figure EV4B and described in the Results section of the revised manuscript (page 9, lines 1–5).

Second, TMEM25 KD facilitated Par3 localization to the TJ following calcium switch, which is consistent with an accelerated TJ development; and TMEM25 KD did not affect correct localization of Par3 to the TJ in fully polarized cells. These results have been presented in Figure EV4A and described in the Results section of the revised manuscript (page 8, lines 8–11 from the bottom).

Minor concerns:

Referee 2 asked the effect of TMEM25 knockdown on macromolecular flux following a calcium switch.

To address this question, we have performed experiments using FITC-dextran and found that decline in paracellular flux of FITC-dextran after a calcium switch is accelerated in TMEM25-depleted cells, supporting the idea that TMEM25 negatively regulates TJ development. The data has been presented in new Figure 3B and described in the Results section of the revised manuscript (page 8, lines 17–20).

Answers to Referee 3

We thank Referee 3 very much for an excellent and thorough review of our manuscript. According to the suggestions raised by Referee 3, we have performed many experiments and revised the manuscript.

Responses to each of points raised by Referee 3 are included below.

We appreciate the referee's help to improve this paper.

We hope that the paper is now suitable for publication in *EMBO Reports*.

Main points:

(1) Referee 3 suggested that the authors should confirm the proposal that TMEM25 attenuates claudin oligomerization or polymerization by claudin1 and claudin2 staining in MDCK cells and in the HEK293 overexpression system. The referee suggested that the authors should answer the question whether co-expression of TMEM25 alters claudin strand formation using the HEK293 overexpression system. To answer these questions, we have performed many experiments and found that ectopically-expressed claudin1 or claudin2 was enriched at contact sites between two claudin-expressing HEK293 cells and the enrichment was effectively suppressed by co-expression of TMEM25, indicating that TMEM25 attenuates claudin trans-interaction and subsequent strand formation. The result has been presented in Figure 6E and described in the Results section of the revised manuscript (page 11, lines 11–27). Furthermore, we have also found that overexpression of TMEM25 in MDCK cells inhibited accumulation of endogenous claudin1 and claudin2 at cell–cell contact regions, which has been presented in Figure EV5B and described in the Results section of the revised manuscript (page 11, lines 24–27).

(2) As pointed out by Referee 3, the role of Par3 remains rather unclear. Hence, the referee suggested that the authors should check whether the localization of TMEM25 is altered by increasing Par3 concentration. According to the suggestion, we have performed experiments using polarized MDCK cells. Under the present experimental conditions, Par3 did not alter the localization of TMEM25 to the TJ, which has been presented in Figure EV5C and described in the Results section of the revised manuscript (page 12, lines 6–8). On the other hand, it is still possible that Par3 sequesters TMEM25 from claudins within the TJ. This possibility has been also described in the Results section of the revised manuscript (page 12, lines 8–9).

Minor points:

(1) In Figure 1D, Referee 3 asked why the higher band is more likely to be pulled down by HA-Par3 and whether glycosylation is important for Par3 binding.

As indicated by the referee, Par3 preferentially binds to the complex-type glycan-bearing TMEM25, compared with the high-mannose glycan-bearing TMEM25, which has been more clearly shown in Figure 1D of the revised manuscript. Because the complex-type glycan-bearing transmembrane proteins are transported to the plasma membrane but the high-mannose glycan-bearing proteins reside at the endoplasmic reticulum, Par3–TMEM25 interaction seems to occur mainly at the plasma membrane. Hence, we have performed experiments and confirmed that Par3 colocalized with TMEM25 at the plasma membrane under the present conditions. This data has been presented in Figure EV2A and the role of glycosylation has been described in the Results section of the revised manuscript (page 6, lines 4–9).

(2) Referee 3 kindly pointed out a typo in the introduction, line 4: adherence junction should be adherens junction.

We have corrected this error in the revised manuscript (page 3, line 5).

Dear Prof. Sumimoto,

Thank you for submitting your revised manuscript. It has now been seen by all of the original referees.

My colleague Ioannis has moved on from EMBO Reports. I have thus stepped in as the handling editor of your manuscript.

As you can see, the referees find that the study is significantly improved during revision and recommend publication. However, I need you to address the points below before I can accept the manuscript.

- Please remove the Author Contribution section from the manuscript text.
- The figures (main and Expanded View) need to be re-submitted as one file per figure.
- Our data editors have asked you to clarify the below points in the figure legends:
 - o Please note that the box plots need to be defined in terms of minima, maxima, centre, bounds of box and whiskers, and percentile in the legend of figure 6e.
- Papers published in EMBO Reports include a 'synopsis' and 'bullet points' to further enhance discoverability. Both are displayed on the html version of the paper and are freely accessible to all readers. The synopsis includes a short standfirst summarizing the study in 1 or 2 sentences (max 35 words) that summarize the paper and are provided by the authors and streamlined by the handling editor. I would therefore ask you to include your synopsis blurb and 3-5 bullet points listing the key experimental findings.
- In addition, please provide an image for the synopsis. This image should provide a rapid overview of the question addressed in the study but still needs to be kept fairly modest since the image size cannot exceed 550 (width) x 300-600 (height) pixels.

Thank you again for giving us to consider your manuscript for EMBO Reports, I look forward to your minor revision.

Kind regards,

Deniz Senyilmaz Tiebe

--

Deniz Senyilmaz Tiebe, PhD
Editor
EMBO Reports

Referee #1:

The authors have done a great job in addressing all issues and concerns that I raised concerning the original manuscript. The revised manuscript provides very solid and exciting new information on the regulation of TJs. I strongly recommend publication of this work in EMBO Reports.

Referee #2:

New data provided by the authors in the revised version of the manuscript have addressed all my comments. Therefore, I recommend the publication of this manuscript without further revisions in EMBO reports.

Referee #3:

In the revision the authors clarified most of my questions. The binding interactions between PAR3 and TMEM25 unfortunately remain unclear. However, overall the revised manuscript is a convincing and strong paper. I recommend publication without further revision.

All editorial and formatting issues were resolved by the authors.

Dear Prof. Sumimoto,

Thank you for submitting your revised manuscript. I have now looked at everything and all is fine. Therefore, I am very pleased to accept your manuscript for publication in EMBO Reports.

Congratulations on a nice work!

Kind regards,

Deniz Senyilmaz Tiebe

--

Deniz Senyilmaz Tiebe, PhD

Editor

EMBO Reports

--
